



# Mobile MAX-DOAS observations of tropospheric NO₂ and HCHO during summer over the Three Rivers' Source region in China

Siyang Cheng[1,2], Xinghong Cheng[1], Jianzhong Ma[1], Xiangde Xu[1], Wenqian Zhang[1], Jinguang Lv[2], Gang Bai[3], Bing Chen[4], Siying Ma[4], Steffen Dörner[5], Sebastian Donner[5], Thomas Wagner[5]

[1]State Key Laboratory of Severe Weather & Institute of Tibetan Plateau Meteorology, Chinese Academy of Meteorological Sciences, Beijing, 100081, China
[2]State Key Laboratory of Applied Optics, Changchun Institute of Optics, Fine Mechanics and Physics, Chinese Academy of Sciences, Changchun, 130033, China
[3]Beijing Remnant Technology Co.,Ltd, Beijing, 100012, China
[4]College of Electronic Engineering, Chengdu University of Information Technology, Chengdu, 610225, China
[5]Max Planck Institute for Chemistry, Mainz, D-55020, Germany

*Correspondence to*: Jianzhong Ma (majz@cma.gov.cn) and Xinghong Cheng (cxingh@cma.gov.cn)

**Abstract.** The tropospheric concentrations of nitrogen dioxide (NO₂) and formaldehyde (HCHO) have high spatio-temporal variability, and in situ observations of these trace gases are still scarce especially in remote background areas. We made four similar circling journeys of mobile MAX-DOAS measurements in the Three Rivers' Source region over the Tibetan Plateau in summer (18–30 July) 2021 for the first time. The differential slant column densities (DSCDs) of NO₂ and HCHO were retrieved from the measured spectra with very weak absorptions along the driving routes. The tropospheric NO₂ and HCHO vertical column densities (VCDs) were calculated from their DSCDs by the geometric approximation method, and they were further filtered to form reliable data sets by eliminating the influences of sunlight shelters and vehicle's vibration and bumpiness. The observational data show that the tropospheric NO₂ and HCHO VCDs decreased with the increasing altitude of the driving route, whose background levels were $0.40 \times 10^{15}$ molec cm⁻² for NO₂ and $2.27 \times 10^{15}$ molec cm⁻² for HCHO in July 2021 over the Three Rivers' Source region. The NO₂ VCDs show similar geographical distribution patterns between the different circling journeys, but the levels of the HCHO VCDs are different between the different circling journeys. The NO₂ VCDs tended to have peak values in the early morning and late afternoon, while the diurnal variation pattern of HCHO VCDs changed with the driving route. The elevated NO₂ VCDs along the driving routes were usually corresponding to enhanced transport emissions from the towns crossed. However, the spatial distributions of the HCHO VCDs depended significantly on natural and meteorological conditions, such as surface temperature. By comparing TROPOMI satellite products and mobile MAX-DOAS results, we found that TROPOMI is still unable to identify the fine-scale spatial variability in tropospheric NO₂ and HCHO VCDs in the background atmosphere over the Tibetan Plateau. Our study provides valuable data sets and information of NO₂ and HCHO over the Tibetan Plateau, benefitting the scientific community in investigating the evolution of atmospheric composition with high spatio-temporal resolution in the background atmosphere at high altitudes, validating and improving the satellite products over mountain terrains, and evaluating the model's ability in simulating atmospheric chemistry over the Tibetan Plateau.



## 1 Introduction

The Tibetan Plateau, also known as the Qinghai-Tibet Plateau in China, is usually called as "the Third Pole" (or "the Roof of the World") with an average surface altitude of 4000~5000 m, covering a vast region located at 73~105 °E longitude and 26~40 °N latitude (Qiu, 2008). Due to thermal and dynamic processes on the role of high altitude and large terrain, the Tibetan Plateau has an important influence on the atmospheric circulation (such as Asian Summer Monsoon), Asian climate and even global climate, and hydrological cycle (Bolin, 1950; Boos and Kuang, 2010; Dong et al., 2017; Duan et al., 2007; Liu et al.,

2007; Yanai et al., 1992; Zhou et al., 2009). As the "Asian Water Towers", there are many water resources in the forms of glaciers, snow packs, lakes and rivers over the Tibetan Plateau, which is the headwaters of major rivers in Asia (such as Indus River, Ganges River, Yangtze River, Yellow River and Lancang River) and influences the economic development and billions of people survival in the downstream region (Xu et al., 2008; Gao et al., 2019). Therefore, the area of "Three Rivers' Source" (i.e. Yangtze River, Yellow River and Lancang River) was established as one of the first five national parks in China in 2021

to better protect the ecological environment. However, we still know very little about the ecological environment including atmospheric environment over this region at present. Almost no observations focus on the abundances and variations of atmospheric composition over the Three Rivers' Source region, limited by the grinding environment, extremely high altitude, topographical heterogeneity, variable weather, and the sparseness of effective techniques and methods. As one of the remote regions in Eurasia, the Tibetan Plateau with low anthropogenic activities and a low population density can be considered as

natural laboratory to investigate the background atmospheric chemistry of the inner Eurasian continent (Ma et al., 2021). With increasing emissions of air pollution over the Tibetan Plateau and its surrounding areas (such as tourism in summer), measurements of the background atmosphere with high spatial-temporal resolution are urgent to improve the understanding of the spatio-temporal evolution of the atmospheric composition (Singh, 2021; Yang et al., 2019; Kang et al., 2022).

Nitrogen dioxide ($NO_2$) and formaldehyde (HCHO) are two important traces gases in the troposphere, participating in the

control of the strong atmospheric oxidant of ozone ($O_3$) (Seinfeld and Pandis, 2016). Nitrogen oxides ($NO_x$), i.e. the sum of $NO_2$ and nitric oxide (NO), not only can be released by various anthropogenic emission sources, such as the burning of fossil fuel and biomass, but also can be emitted by natural processes including microbial activities in soils and lightning in the atmosphere (Lee et al., 1997; Granier et al., 2011; Kurokawa et al., 2013). HCHO is produced not only by primary sources (e.g. emissions of industry and transportation in city and biomass burning), but also by photochemical oxidation of methane

and non-methane volatile organic compounds (e.g. isoprene emitted from natural vegetation) in the remote atmosphere (Stavrakou et al., 2009). High-accuracy measurements of $NO_2$ and HCHO with high spatial and temporal resolution are beneficial to understand their variation characteristics in the background atmosphere, quite useful to validate the satellite products, and very valuable to explore processes of atmospheric chemistry.

The ground-based observations of $NO_2$ and HCHO concentrations in the background atmosphere at high altitude are

relatively scarce at present. Under the frameworks of the Global Atmosphere Watch program of the World Meteorological Organization (WMO/GAW) and the Network for the Detection of Atmospheric Composition Change (NDACC), long-term



observations of atmospheric composition have been carried out at some high mountain stations, such as the Waliguan (WLG; 3816 m above sea level) global atmosphere background observatory, located in the northeastern part of the Tibetan Plateau (Xu et al., 2020; Ma et al., 2021). With respect to $NO_2$ at WLG, previous studies found different levels (5~600 ppt) of $NO_2$

during different periods, leading to a positive or negative sign of net ozone production in the remote troposphere (Xue et al., 2011; Meng et al., 2010; Ma et al., 2020; Ma et al., 2002). Short-term HCHO observations at WLG in 2005 indicated that the possible sources for HCHO were photo-oxidation of biogenic emission of isoprene, animal excrement, and long-distance transportation from polluted air (Mu et al., 2007). The two stations of Qinghai Lake and Menyuan are adjacent to WLG, but the diurnal variations of $NO_x$ ($NO_2$) are different and possibly influenced by traffic and residential emissions, complex terrain,

boundary layer processes, and transport from city air masses (Wang et al., 2015; Zhao et al., 2020). According to the measurements at the Qomolangma Atmospheric and Environmental Observation and Research Station (QOMS; 4276 m above sea level) of the south-central Tibetan Plateau from December 2017 to March 2019, the levels of $NO_2$ and HCHO were significantly higher than those at WLG station, related to local emissions (e.g. tourism, biomass burning, vegetation) and air pollution transport from the South Asia (Xing et al., 2021; Ma et al., 2020). Increased concentrations of tropospheric $NO_2$ at

QOMS are concentrated in the lower layers with obvious seasonal variations (peak of 1.28 ppb in autumn) and diurnal variations (two peaks at 11:00~13:00 BJT and after 16:00 BJT) (Xing et al., 2021). The tropospheric HCHO vertical distribution showed an exponential shape at QOMS with a seasonal peak of 5.20 ppb in autumn, and the peaks of HCHO appeared between 10:00~16:00 BJT in winter and spring and after 16:00 BJT in summer and autumn, respectively (Xing et al., 2021). In recent years, the China National Environmental Monitoring Center (CNEMC) also established several

atmospheric composition monitoring stations over the Tibetan Plateau, but they mainly focused on the continuous monitoring of the surface particulate matter with aerodynamic diameter less than 2.5 μm and 10 μm ($PM_{2.5}$ and $PM_{10}$), $NO_2$, sulphur dioxide ($SO_2$), $O_3$, and carbon oxide (CO) in cities, such as Lhasa and Xining (Chen et al., 2019; He et al., 2017; Yang et al., 2019). As a whole, these station observations cannot meet the demand of detecting the $NO_2$ and HCHO variations with high spatial resolution over the Tibetan Plateau, which are also crucial to the validation of satellite products over areas with complex

terrain. To the best of our knowledge, there are no reports about mobile measurements of $NO_2$ and HCHO in the background atmosphere over the Tibetan Plateau.

The measurements of $NO_2$ and HCHO with high spatial and temporal resolution are challenging over the Tibetan Plateau. In the early days, some studies on $NO_2$ and HCHO were based on the time-consuming air sampling method (Mu et al., 2007; Meng et al., 2010; Ma et al., 2002). With the development of measurement techniques, in-situ methods started to be applied

to measure surface $NO_x$ ($NO_2$) and HCHO concentrations at a few stations (Wang et al., 2006; Xue et al., 2011; Wang et al., 2015; Zhao et al., 2020; Ran et al., 2014; Chen et al., 2019; Yang et al., 2019; Xue et al., 2013; Duo et al., 2018). However, there are limitations in the spatio-temporal representation for the sampling and in-situ measurement methods. As an alternative, satellite remote-sensing can perform long-term observations of $NO_2$ and HCHO and cover large areas with sparse spatio-temporal resolution, but the uncertainties of satellite $NO_2$ and HCHO products are rather large owing to complex terrain and

weather over the Tibetan Plateau (Guo et al., 2016; Zhang et al., 2021). As a kind of advanced ground-based remote-sensing





technique, Multi-AXis Differential Optical Absorption Spectroscopy (MAX-DOAS) has been certified in the measurement techniques of NDACC (Mazière et al., 2018). The successful observations of trace gases with very low abundances by MAX-DOAS depend on multi-factors, such as long optical paths, a high signal-to-noise ratio of the instrument, and characteristic spectral absorption features of the target species. According to previous studies, MAX-DOAS has the potential to measure

tropospheric trace gases with very low level mixing ratios (ppt order for $NO_2$ and sub-ppb order for HCHO) in the background atmosphere at high altitude stations (Franco et al., 2015; Gil-Ojeda et al., 2015; Gomez et al., 2014; Marais et al., 2021; Schreier et al., 2016). Also, this technique has been used to measure the levels and monthly variations of $NO_2$ and HCHO in the global pristine atmosphere at WLG station (Ma et al., 2020). Stratospheric $O_3$ and its depleting substances (including $NO_2$) have been successfully retrieved from zenith DOAS spectra at a clean suburb station in the northern Tibetan Plateau (Cheng et al., 2021).

Moreover, ground-based MAX-DOAS has been applied to monitor vertical distributions of $NO_2$ and HCHO in the southern Tibetan Plateau (Xing et al., 2021). Comparing with MAX-DOAS observation at a fixed site, mobile MAX-DOAS measurements in the background atmosphere over the Tibetan Plateau is a greater challenge, because: (1) Vehicle's violent vibration and bumpiness reduce the stability of the signal acquisition and even introduce unknown interference signals; (2) The measured signals can be strongly reduced by shelters due to complex terrain, such as tunnels, bridges, signposts, and

mountains (usually such measurements have to be filtered out); and (3) The observations in practice are also controlled by various factors, e.g. variable weather, hypoxic environment in the plateau, geospatial signal loss and problems with the power supply. Therefore, the $NO_2$ and HCHO concentrations measured by mobile MAX-DOAS over the Tibetan Plateau are an extremely valuable data sets to characterise the spatio-temporal evolution of the atmospheric composition in the background atmosphere, validation and improvement of satellite products over mountain terrain, and evaluation of the simulation results

of atmospheric chemistry models over the Tibetan Plateau.

We made the mobile MAX-DOAS measurements in July 2021 over the plateau terrain for the first time. In this study, the primary objective is to analyse the spectra of scattered sun light collected in the Three Rivers' Source region over the Tibetan Plateau, obtain the data sets of tropospheric $NO_2$ and HCHO vertical column densities (VCDs) in the background atmosphere with high spatio-temporal resolution, and investigate the abundances and spatio-temporal variations of tropospheric $NO_2$ and

HCHO VCDs during the field campaign. Large effort was spent on the spectral analysis and data filtering to obtain reliable tropospheric $NO_2$ and HCHO VCDs, because of the very weak spectral absorptions of the respective trace gases in the background atmosphere at high altitude as well as the influences of shelters and vehicle's vibration and bumpiness along the driving routes. In Section 2, we describe the field experiment in July 2021 over the Tibetan Plateau, including the observation vehicle, MAX-DOAS instrument, experiment region and deployment strategies. Section 3 introduces the spectral analysis as

well as the calculation and filtering of the $NO_2$ and HCHO VCDs. In Section 4, we present the abundances, temporal variation and spatial distribution of the tropospheric $NO_2$ and HCHO VCDs during the field campaign, as well as the comparison with TROPOMI products. Summary and conclusions are given in Section 5.



## 2 Field experiment

### 2.1 Description of vehicle and instrumentation

A mobile vehicle has been designed and assembled for measurements of atmospheric composition over the Tibetan Plateau (Fig. 1a). The mobile vehicle has been operated by the Chinese Academy of Meteorological Sciences (CAMS) since February 2021. The outside parts of instrumentation are fixed on the roof of the vehicle, which is about 3.5 m above the ground. The outside parts of instrumentation contain the sensors for spatial position (longitude, latitude, altitude) and attitude (yaw, pitch and roll angles) of the mobile vehicle. The units of the system control, data collection, screen display and Uninterruptible

Power Supply (UPS) are mounted in the interior of the mobile vehicle. The UPS's battery pack, recharged after the mobile vehicle reaches the destination of observation route, can offer operation time of around 16 h with a power of 2000 W. All instrumentations have been specially reinforced to allow the mobile vehicle to travel over the complex road conditions of the Tibetan Plateau. The mobile vehicle usually runs at a speed of ~60 km/h for motorways and ~40 km/h for ordinary roads, respectively, during our field experiment.

For the field campaign of mobile observations of the atmospheric environment over the Tibetan Plateau, the aforementioned vehicle was equipped with an instrument called Tube MAX-DOAS, developed by the Max Planck Institute for Chemistry (MPIC), Mainz, Germany. The Tube MAX-DOAS contained two parts, one outside (Fig. 1b) and another inside (Fig. 1c) the vehicle, respectively. (1) The outside part was fixed on the rear of the vehicle's roof and is mainly composed of the telescope, optical fibre, stepper motor, tubular shell, and protective cover. The telescope, pointing to the back of the vehicle, rotated in

the vertical plane to achieve the measurement at seven different elevation angles (3°, 6°, 10°, 15°, 20°, 30°, 90°) driven by the stepper motor. The scattered sunlight was collected by the telescope and transferred to a spectrograph inside the vehicle via the optical fibre. (2) The inside part was made up of the spectrograph, data collection unit, temperature control unit as well as a laptop which controls the instrument operation and data collection. For each elevation angle, the Tube MAX-DOAS collected one spectrum at a stable detector temperature of $15 \pm 0.1$ ℃ with the integration time of ~1 min. The spectrograph covered the

wavelength range of 300~466 nm with ~0.6 nm spectral resolution. The Tube MAX-DOAS not only automatically collected the scattered sunlight spectra for the cyclic elevation angle sequences during daytime, but also recorded spectra of dark current (DC) and electronic offset (OS) at night for correcting the daytime spectra of scattered sun light. The laptop coordinated the operation of each module during the measurement procedure. The MPIC Tube MAX-DOAS system has been successfully applied to the ground-based observations of atmospheric composition at the Golmud station over the Northern Tibetan Plateau

(Cheng et al., 2021).

### 2.2 Description of the measurement location and deployment strategies

The mobile field observation campaign was performed on the northeast of the Tibetan Plateau in western China (Fig. 2a). With the average altitude of 3000 ~ 5000 m, this region is the source catchment area of many rivers, such as the Yangtze River, Yellow River and Lancang River (i.e. so-called "Three Rivers' Source"). The Three Rivers' Source region is also one of the



regions with the highest concentration of high-altitude biodiversity in the world. As one of five national parks, the establishment of the Three Rivers' Source national park was approved by the China State Council on 30 September 2021. The main vegetation types are alpine steppe and meadows in the region along the observation route, belonging to a unique and typical alpine ecosystem. The main landform is the mountain plain in the field measurement area. The Three Rivers' Source region has a typical plateau continental climate, characterized by distinct dry and wet seasons, alternate hot and cold seasons,

a small annual temperature difference, a large diurnal temperature difference, long sunshine time, strong solar radiation and four indistinct seasons. There are rapid spatial and temporal variations of the local climate over the Three Rivers' Source region. Yak and sheep grazing in summer is the main industry over the Three Rivers' Source region, isolated from industrial and population centres. Therefore, this remote region is an excellent natural laboratory to investigate the background atmosphere.

In order to reveal the background abundance and spatio-temporal variation of the atmospheric composition over the Three Rivers' Source region, we took various factors into consideration during the design of the deployment strategies, such as the regional representativeness of the driving routes, the technical requirement of the passive MAX-DOAS measurement, the sunlight shelter and the bumpy condition along the driving route, the reliable electric power safeguard, and first aid for sudden altitude sickness. Finally, the mobile MAX-DOAS field experiment was carried out in the southeast of Qinghai province,

China (Fig. 2b). The driving routes traverse the Yangtze River and the Yellow River and are close to the Lancang River. It took three days for one circling journey. Four circling journeys were made during the mobile MAX-DOAS field experiment period in July 2021 (Table 1). We drove from the meteorological bureau of Xining city, the capital of Qinghai province, to the meteorological bureau of Dari county of the Guoluo Tibetan autonomous prefecture, south-eastern Qinghai province, on the first day of each circling journey (red curve in Fig. 2b). We travelled from the meteorological bureau of Dari county to the

meteorological bureau of the Yushu Tibetan autonomous prefecture on the middle day (blue curve in Fig. 2b). We returned to Xining city from the Yushu Tibetan autonomous prefecture on the third day (black curve in Fig. 2b). Hereafter the three segments of the closed-loop journey are referred to as XD, DY, YX, respectively. The durations were about 12 h, 8 h, and 13 h for the XD, DY, and YX segments, respectively. Most of driving routes are motorways, except parts of the national roads in the YX segment. More sunlight shelters occurred in the XD segment, because of the tunnels, bridges, signposts, and mountains.

The observed MAX-DOAS data were saved in the laptop, and backed up when arriving at the terminus of each segment of the journey. In addition to troubleshooting by field observers, our MAX-DOAS team also provided the technical support via remote wireless network during the campaign.

## 3 Spectral retrieval and data filtering

### 3.1 Spectral analysis

Based on the Beer-Lambert law, the column densities of trace gases can be retrieved from the scattered sunlight spectra by the widely used method of Differential optical absorption spectroscopy (DOAS) (Platt and Stutz, 2008). The basic idea of DOAS



is to decompose the atmospheric spectral extinction into two terms, i.e. terms with slow spectral variation (such as atmospheric scattering) and fast variation (mainly trace gas absorptions) with wavelength. The slant column density (SCD) of a trace gas is defined as its concentration integrated along the effective light path. The total (from the instrument to the top of atmosphere)

SCD can be split into two parts, i.e. so-called tropospheric SCD ($SCD_{Trop}$) and stratospheric SCD ($SCD_{Stra}$). For species concentrated in the troposphere or light traversing the same path in the stratosphere for different elevation angles ($\alpha$), the $SCD_{Stra}$ can be neglected or cancels out, which means $SCD_{\alpha,Stra} \approx SCD_{90,Stra}$. In the practice of the MAX-DOAS spectral analysis, a Fraunhofer reference spectrum (FRS) needs to be selected to correct the strong solar Fraunhofer lines. Thus the result of the spectral analysis is the so-called differential slant column density (DSCD) of the target species (such as $NO_2$ and

HCHO in this study), which represents the difference in trace gas absorption between the measured atmospheric spectrum and the FRS. There are two schemes for the FRS selection from measured spectra: one is using a fixed spectrum (hereafter named "fixed FRS"), usually at the 90°elevation angle during noon to minimize the tropospheric and stratospheric contributions, for all measured spectra; the other is using sequential spectra (hereafter named "sequential FRS"), which are defined as the interpolated spectra between two zenith spectra measured before and after an off-zenith sequence of elevation angles. Due to

more similar atmospheric conditions and instrument properties between a specific measured spectrum and the corresponding sequential FRS, higher signal-to-noise ratios and smaller fitting errors are achieved by using a sequential FRS than a fixed FRS. Fig. 3 shows the root mean square (RMS) of the spectral fitting residuals using a fixed FRS and sequential FRS for $NO_2$ and HCHO, respectively. It is clear that the RMS medians are smaller for using a sequential FRS than that for a fixed FRS. Thus we prefer to use the sequential FRS for the mobile MAX-DOAS measurement in this study. For $NO_2$, we can retrieve

the DSCD not only in the ultraviolet (UV) region (351~390 nm) but also in the visible region (400~434 nm). Fig. 4 compares the $NO_2$ DSCDs and the RMSs of the spectral fitting residuals for using either the visible and UV spectral interval. The overall trends of the $NO_2$ DSCDs are consistent between both spectral intervals with the correlation coefficient of R=0.75, but the averaged RMSs of the spectral fitting residuals in the visible wavelength region, i.e. $(6.26 \pm 6.92) \times 10^{-4}$, are smaller than those in the UV wavelength interval, i.e. $(7.62 \pm 9.17) \times 10^{-4}$. The final settings of the $NO_2$ and HCHO spectral retrieval parameters,

such as cross sections of the target and interference species, Ring spectra, polynomial degree and intensity offset, similar as in previous studies (Cheng et al., 2022; Cheng et al., 2019), see Table 2. The spectral analysis, including DC and OS corrections of the measurement spectra and the spectral calibration of the FRS, was implemented by the QDOAS software based on a non-linear least squares fitting method, developed by the Royal Belgian Institute for Space Aeronomy (BIRA-IASB) (Danckaert et al., 2017). Fig. 5 shows an example of the spectral fitting for the $NO_2$ and HCHO DSCDs from a spectrum measured at the

elevation angle of 15° at 11:02 BJT on 18 July 2021 (SZA = 34.11°). In the post processing of $NO_2$ and HCHO DSCDs, we applied the following filters: RMS < 0.005; offset (constant) should be between ± 0.03; SZA < 80 °. These filters were selected as they provide a good balance between quality of the results and skipping not too many data. These filters almost filtered out all "bad measurements", which were caused by sunlight shelters and bumpy conditions. Finally, relative to measurements with SZA<90 °, the percentages of remaining DSCD data were 69% for $NO_2$ and 74% for HCHO, respectively.



## 3.2 NO₂ and HCHO VCDs

Based on the aforementioned filtered NO$_2$ and HCHO DSCDs retrieved from the spectra, we need to firstly obtain the tropospheric DSCDs at the elevation angle $\alpha$ (i.e., $\mathrm{DSCD}_{\alpha,\mathrm{Trop}} \equiv \mathrm{SCD}_{\alpha,\mathrm{Trop}} - \mathrm{SCD}_{90,\mathrm{Trop}}$), which are used to calculate the NO$_2$ and HCHO vertical column densities (VCDs) in the troposphere. In the situation of fixed FRS, the $\mathrm{DSCD}_{\mathrm{Trop}}$ are produced by the DSCDs of off-zenith viewing direction minus that at 90 ° elevation angle of the same elevation sequence; In the case of sequential FRS in this study, the DSCDs from spectral inversion can be regarded as $\mathrm{DSCD}_{\mathrm{Trop}}$ (Hönninger et al., 2004).

The SCDs (or DSCDs) depend on the concentration profile of target species, effective light path length, measurement geometry and solar position. Using the air mass factor (AMF), the SCDs (or DSCDs) can be converted to the VCDs, which are independent of the light path and the observation geometry and thus convenient for comparison between different measurements. The tropospheric AMF at the elevation angle $\alpha$ ($\mathrm{AMF}_{\alpha,\mathrm{Trop}}$) is given by the ratio of the SCD to VCD in the troposphere:

$$\mathrm{AMF}_{\alpha,\mathrm{Trop}} = \frac{\mathrm{SCD}_{\alpha,\mathrm{Trop}}}{\mathrm{VCD}_{\mathrm{Trop}}} \tag{1}$$

If $\alpha = 90°$,

$$\mathrm{AMF}_{90,\mathrm{Trop}} = \frac{\mathrm{SCD}_{90,\mathrm{Trop}}}{\mathrm{VCD}_{\mathrm{Trop}}} \tag{2}$$

We define the $\mathrm{DAMF}_{\alpha,\mathrm{Trop}}$ as the tropospheric differential AMF, i.e.

$$\mathrm{DAMF}_{\alpha,\mathrm{Trop}} = \mathrm{AMF}_{\alpha,\mathrm{Trop}} - \mathrm{AMF}_{90,\mathrm{Trop}} \tag{3}$$

By equation (1) minus equation (2), $\mathrm{VCD}_{\mathrm{Trop}}$ can be deduced:

$$\mathrm{VCD}_{\mathrm{Trop}} = \frac{\mathrm{SCD}_{\alpha,\mathrm{Trop}} - \mathrm{SCD}_{90,\mathrm{Trop}}}{\mathrm{AMF}_{\alpha,\mathrm{Trop}} - \mathrm{AMF}_{90,\mathrm{Trop}}} = \frac{\mathrm{DSCD}_{\alpha,\mathrm{Trop}}}{\mathrm{DAMF}_{\alpha,\mathrm{Trop}}} \tag{4}$$

where the AMF can be simulated by an atmospheric radiative transfer model or estimated by the method of geometric approximation. The former method is more exact, but depends on various input parameters, such as the profiles of trace gas and aerosol, which are usually not known. The latter method is less correct, but relatively simple and also applicative to get the $\mathrm{VCD}_{\mathrm{Trop}}$, if target species are mainly distributed in the lower troposphere. Due to the lack of necessary data over the Tibetan Plateau to simulate the correct NO$_2$ and HCHO AMFs, we adopted the geometric approximation method in this study. Here it should be noted that the errors caused by the geometric approximation method are much smaller for measurements at high altitudes, because the scattering probability is much smaller compared to measurements at sea level. Thus the direct viewing path length becomes longer and is in better agreement with the assumptions of the geometric approximation method. The $\mathrm{AMF}_{\alpha,\mathrm{Trop}}$ in the condition of geometric approximation can be expressed as:

$$\mathrm{AMF}_{\alpha,\mathrm{Trop}} \approx \frac{1}{\sin(\alpha)} = \sin^{-1}(\alpha) \tag{5}$$

Therefore, equation (4) becomes

$$\mathrm{VCD}_{\mathrm{Trop}} = \frac{\mathrm{DSCD}_{\alpha,\mathrm{Trop}}}{\sin^{-1}(\alpha) - 1}, (\alpha \neq 90°, \mathrm{AMF}_{90,\mathrm{Trop}} = 1) \tag{6}$$





Ideally, the elevation angles should be corrected by the attitude angles of the mobile vehicle when applying the geometric approximation. However, the partial system of the attitude angles of the mobile vehicle did not work well, which may be connected with the special environment of Tibetan Plateau (such as low atmospheric pressure) and bumpiness of the mobile observation platform (leading to instabilities of the data collection). Thus we use the uncorrected elevation angles during the conversion of DSCD to VCD in equation (6). Of course, the uncorrected elevation angles will cause some errors if the mobile

observation vehicle is not on a horizontal surface, but on average these errors will cancel out. Also these errors are typically small for the larger elevation angles (for example, $15°$, $20°$, $30°$) and can be neglected when compared to other uncertainties. To further judge how good the geometric approximation is, the resulting VCDs derived for different elevation angles have been compared. Fig. 6 shows the $NO_2$ (HCHO) VCDs between the three elevation angles ($15°$, $20°$, $30°$). The VCDs are rather consistent at the three elevation angles with correlation coefficients of R=0.91~0.95 for $NO_2$ and R=0.66~0.80 for HCHO,

respectively (Table 3). This implies that the geometric approximation method has high accuracy. The standard deviation of the $NO_2$ (HCHO) VCDs is small at $15°$ elevation angles (Fig. 6), implying the high reliability of VCDs at $15°$ elevation angle ($VCD_{15°}$). Therefore, to compromise between accuracy of the geometric approximation and signal to noise, the $VCD_{15°}$ were treated as the reliable results on a selection criterion (for $NO_2$, the absolute difference of VCDs between $15°$ and $20°$ is < $1\times10^{15}$ molec cm$^{-2}$ or the relative difference is <5%; for HCHO, the absolute difference of VCDs between $15°$ and $20°$ is <

$2\times10^{15}$ molec cm$^{-2}$ or the relative difference is <5%). The filtered $NO_2$ and HCHO $VCD_{15°}$ during the mobile measurement period were kept as the final results to explore the background abundance and spatio-temporal variation of $NO_2$ and HCHO over the Three Rivers' Source region of the Tibetan Plateau.

## 4 Interpretation of the results

### 4.1 Abundance

Based on filtered final $NO_2$ and HCHO VCDs, the means ± standard deviations were 0.69 ± 1.13 $\times10^{15}$ molec cm$^{-2}$ for $NO_2$ and 2.43 ± 1.66 $\times10^{15}$ molec cm$^{-2}$ for HCHO in July 2021 along the driving routes. The background levels of $NO_2$ and HCHO VCDs can be estimated by the maximum frequency method (Cheng et al., 2017). According to the Lorentz fitted curves of the relative frequency distribution of the $NO_2$ and HCHO VCDs during the field campaign (Fig. 7a), the background levels were 0.40 $\times10^{15}$ molec cm$^{-2}$ for $NO_2$ and 2.27 $\times10^{15}$ molec cm$^{-2}$ for HCHO in summer on the northeast of the Tibetan Plateau.

These values are smaller than those observed in summer 2018 at the Qomolangma Atmospheric and Environmental Observation and Research Station of the Chinese Academy of Sciences, located in the south-central Tibetan Plateau (medians of 0.80 $\times10^{15}$ molec cm$^{-2}$ for $NO_2$ and 3.13 $\times10^{15}$ molec cm$^{-2}$ for HCHO, respectively) (Xing et al., 2021). To explore the dependence of the $NO_2$ and HCHO VCDs on the route altitude (in the range of 2280~4830 m), we divided the mobile route altitudes into vertical bins with intervals of 500 m. Fig. 7b shows the means and standard deviations of the $NO_2$ and HCHO

VCDs in each vertical grid cell. There are generally decreasing trends with increasing altitude. This is consistent with our





knowledge of the natural background atmosphere, i.e. the higher the altitude, the lower the air density. Different from the nearly constant decreasing rate of the HCHO VCDs with the route altitude, there are at least two segments with significantly different decreasing rates above and below 2750 m altitude. The $NO_2$ VCDs in the 2000~2500 m grid cell (8.17 $\times10^{15}$ molec $cm^{-2}$) were substantially larger because the mobile route was close to the city of Xining (about 2260 m altitude), where

there are stronger anthropogenic emission sources of air pollutants, such as increased urban transport emissions leading to higher $NO_2$ levels. The $NO_2$ VCDs were quite low in the altitude above 3500 m, partly related to almost no human activities at this altitude. Due to very limited emissions of anthropogenic volatile organic compounds (VOCs) over the Tibetan Plateau, the changes of the HCHO VCDs with altitude were likely to be primarily connected with the natural process, such as the oxidation of methane and non-methane volatile organic compounds (Stavrakou et al., 2009). As a whole, the measurements

(except close to the cities) at the higher altitudes in summer are able to reflect the background atmosphere with rather low $NO_2$ and HCHO levels over the Three Rivers' Source region.

## 4.2 Temporal variation

The daily variations of the $NO_2$ and HCHO VCDs are shown in Fig. 8. The daily variations of $NO_2$ are similar between different circling journeys, characterized by the larger means and 90th percentiles on the first and the third days (i.e. on the days of the

XD and YX driving routes) and correspondingly lower values on the second day (i.e. on the day of the DY driving route) of each circling journey. The $NO_2$ means are always larger than the medians on each day, especially in the situation of the XD driving route, partly because the driving route covers small areas with very high $NO_2$ abundances, such as Xining city, and large background areas with relatively low $NO_2$ abundances in the XD driving route. For the same driving route of the four circling journeys, the daily $NO_2$ levels are close to each other, with the $NO_2$ medians in the range of 0.19~0.63 $\times10^{15}$

molec $cm^{-2}$ during the field campaign. However, the daily variations of the HCHO VCDs are different from those of $NO_2$. The means and medians of the daily HCHO VCDs are basically consistent on all days, with the maximum mean of 4.63 $\times10^{15}$ molec $cm^{-2}$ on 21 July 2021 and the minimum mean of 1.15 $\times10^{15}$ molec $cm^{-2}$ on 27 July 2021. There are obvious differences in the levels of HCHO VCDs between the different circling journeys. The higher and lower HCHO VCDs appeared during the second circling journey (i.e. 21-23 July 2021) and the third circling journey (i.e. 25-27 July 2021), respectively. HCHO has

large natural vegetation sources, with the emission strength depending strongly on weather conditions such temperature and solar radiation at the Earth's surface (Borovski et al., 2014). We looked at air temperature at 2 m above the land surface and the downward solar radiation at the surface (SSRD), which are derived from hourly ERA5 reanalysis data with 0.25 °×0.25 ° resolution, and interpolated them to the geographical and time intervals corresponding to the daily HCHO VCDs. It is shown that the daily variations between air temperature and HCHO VCDs are highly correlated, with the correlation coefficient of

R=0.95 (Fig. 8b, S1). This implies that higher temperatures are connected with more VOCs emitted by vegetation, leading to higher HCHO VCDs. The daily HCHO VCDs are also related to surface solar radiation, but with a smaller correlation coefficient of R=0.27, which is probably caused by the higher variability of local solar radiation over the Tibetan Plateau compared to the temperature. For different segments of the specific circling journey, the relative variability in the 90th



percentiles of the daily HCHO VCDs is smaller than that of $NO_2$, implying that the local cities over the Tibetan Plateau (such

as Xining) along the driving route have less influence on HCHO compared to $NO_2$. Therefore, the remarkable HCHO daily variations are mainly connected with the variable weather over the Tibetan Plateau, which affects the natural emissions of HCHO precursors significantly.

The average daytime diurnal variations of the tropospheric $NO_2$ VCDs during the mobile MAX-DOAS field campaign in July 2021 over the Three Rivers' Source region of the Tibetan Plateau are presented in Fig. 9. The available time period,

confined by the sunshine duration and the distance of the driving routes, is the shortest for the DY driving route. The diurnal cycle of the $NO_2$ VCD means or medians presents high values in the morning and evening and shows lower levels of ~0.38 $\times 10^{15}$ molec $cm^{-2}$ from 12:00 BJT to 17:00 BJT (Fig. 9a). The means of the $NO_2$ VCD are also significantly higher than the corresponding medians before 11:00 BJT with larger standard deviations. The $NO_2$ diurnal variation patterns of the XD, DY, and YX driving routes are different, although the diurnal patterns are rather consistent for different days of the same driving

route (Fig. 9b-d). The $NO_2$ VCDs sharply decreased in the morning during the XD driving route, with larger standard deviations around 16:00 BJT, when the mobile observation vehicle was close to the toll station. For the DY driving route, the $NO_2$ VCDs stayed at the lower level and then slightly increased in the late afternoon. In the situation of the YX driving route, the diurnal pattern of $NO_2$ VCDs was a symmetric "U" shape. It should be noted that the mobile observation vehicle reached the destination of the YX driving route around 22:00 BJT and the lacking $NO_2$ VCDs were due to SZA > 80 °after 20:00 BJT. The

amplitudes of the $NO_2$ diurnal variation as well as the maxima $NO_2$ level among different driving routes were decreasing in the order of the segments XD, YX, and DY. Previous studies at the background station of lower altitude showed a similar "U" shape of the $NO_2$ diurnal variation, which was connected with the higher photolysis rate owing to stronger solar irradiance at noon and for a site location far away from emission sources (Cheng et al., 2019). Compared with observations in summer at the Qomolangma station in the south-central Tibetan Plateau, the daytime $NO_2$ diurnal pattern in this study is more continuous

(although it is hard to compare quantitatively due to large uncertainties of the $NO_2$ diurnal pattern at Qomolangma caused by a lot of missing data) (Xing et al., 2021). We also checked whether the enhanced $NO_2$ VCDs in the morning and evening might be an artefact caused by the effect of stratospheric $NO_2$ on the derived tropospheric $NO_2$ VCD.   In our data analysis (see section 3.1) it is assumed that the stratospheric $NO_2$ absorption is independent on the elevation angle. While this is not exactly true, it is a valid assumption for typical measurement situations in polluted or slightly polluted environments. If, however, the

tropospheric $NO_2$ absorption is very weak, the remaining stratospheric influence might be substantial. We tested this potential influence of the stratospheric $NO_2$ absorption on the retrieved tropospheric $NO_2$ VCD for our measurements, by performing radiative transfer simulations using a stratospheric $NO_2$ profile with a stratospheric $NO_2$ VCD of 4 $\times 10^{15}$ molec $cm^{-2}$. As a result, we found that for SZA < 80 °the introduced $NO_2$ DSCD for an elevation angle of 15 °is < 1 $\times 10^{15}$ molec $cm^{-2}$ (see Fig. S2) thus leading to a maximum artificial $NO_2$ VCD of $3.5 \times 10^{14}$ molec $cm^{-2}$. Moreover, for SZA<80 °, the artificial $NO_2$ VCD

shows almost no SZA dependence. Thus the potential influence of the stratospheric $NO_2$ absorption cannot explain the observed diurnal cycle of the tropospheric $NO_2$ VCD. From these findings we conclude that the $NO_2$ diurnal variations were primarily caused by enhanced pollution in the morning and evening when the mobile observation vehicle was located in or



close to the cities or county town. An additional effect on the diurnal variation is probably caused by the enhanced $NO_2$ photolysis around noon.

The diurnal variation of the tropospheric hourly HCHO VCDs for the entire campaign, each day, and three segments of the circling journey are shown in Fig. 10. With respect to the total means and medians of the HCHO VCDs in the range of 1.92~4.36 $\times 10^{15}$ molec cm$^{-2}$ (Fig. 10a), their diurnal variations are rather consistent during the whole day. They slightly decrease before 10:00 BJT and increase after 18:00 BJT, and also have no significant differences in the standard deviations. However, the diurnal variations of the HCHO VCDs are obviously different both for different days of the same driving route

or among different driving routes (Fig. 10b-d). On average, the diurnal pattern of the HCHO VCDs during the XD driving route presents a weak "U" shape, i.e. slightly higher HCHO levels in the morning and evening. For the DY driving route, the total averaged HCHO VCDs almost maintain the level around 2 $\times 10^{15}$ molec cm$^{-2}$ before 14:00 BJT, and then gradually increase until the end of the DY journey. The diurnal pattern of the HCHO VCDs for the YX driving route presents a "W" shape, i.e. higher HCHO VCDs occur around 13:00 BJT, in the morning and in the evening. The variable diurnal cycles of

HCHO VCDs were also found by ship-based MAX-DOAS measurements over the middle and lower Yangtze River in winter, where the both primary sources and photochemical secondary formation have large influences (Hong et al., 2018). The daytime HCHO diurnal patterns in this study are also distinguished from those observed by ground-based MAX-DOAS at the Qomolangma station in the south-central Tibetan Plateau in summer, where HCHO peaks appeared around 12:30, 15:00, and 18:00 BJT (Xing et al., 2021). Even at the starting and ending points of the driving route, there were almost no strong HCHO

primary sources caused by anthropogenic activities over the Three Rivers' Source region. Thus we infer that the variable diurnal patterns of HCHO were mainly connected with the secondary photochemical formation of active VOCs emitted from vegetation (Mu et al., 2007). Meanwhile, due to the varying local microclimates over the Tibetan Plateau as well as different types and amounts of vegetation at different altitudes, the temporal variations of secondary HCHO production are quite changeable. More comprehensive observations are needed over the Tibetan Plateau in the future to deeply understand the

HCHO temporal evolution.

### 4.3 Spatial distribution

Figure 11 shows the spatial distributions of the tropospheric $NO_2$ VCDs along the XD, DY, and YX driving routes in July 2021. For the same segment of four circling journeys (i.e. XD, DY, or YX), the tropospheric $NO_2$ VCDs present a nearly consistent spatial distribution. It is also clear that the tropospheric $NO_2$ VCDs were elevated when the mobile observation

vehicle passed through counties or cities, such as Xining and Yushu. This can be attributed to increased anthropogenic activities in cities or counties, such as traffic and residential emissions. There are significantly larger $NO_2$ VCDs on the driving routes of south-eastern Qinghai Lake, which is a famous tourist destination. Moreover, as one of the arterial roads to Tibet, there are many diesel vehicles passing through the basin of Qinghai Lake via national highways surrounding the lake. The touring buses or cars as well as the cargo transport vehicles could lead to the higher $NO_2$ abundances in summer around the Qinghai Lake.

According to previous studies at the northwest section of the Qinghai Lake shore in October of 2010 and 2011, the emissions




from diesel vehicles around Qinghai Lake were likely the main source of nitrogen oxides ($NO_x$) (Wang et al., 2015). The enhanced $NO_2$ levels could even be found at the highway junction (such as the location of 98.97°E, 35.20°N) and the tunnel exit (such as the location of 99.40°E, 34.92°N; Note: The telescope of the MAX-DOAS pointed to the backward of the driving direction) (Fig. 11a1, d1). This situation would not appear once vehicle flowrate was less at these special locations (Fig. 11b1,

c1). The $NO_2$ spatial distributions over the main area of the Three Rivers' Source, such as around the counties of Dari, Shiqu, Chenduo, and Maduo during the DY driving route and the first half of the YX driving route, were relatively uniform with very low levels ($<1 \times 10^{15}$ molec cm$^{-2}$). Previous investigations of the tropospheric ozone chemical budget, simulated and constrained by measured $NO_2$ concentration at the Waliguan background station located in the north-eastern Tibetan Plateau, showed that the $NO_x$ levels play the vital role in the net sign of ozone production from formation and loss reaction for the

tropospheric background atmosphere (Ma et al., 2002; Ma et al., 2020; Xue et al., 2013). Therefore, with the more and more anthropogenic activities, the effects of increasing $NO_2$ levels on the photochemistry and oxidation capacity of the background atmosphere should be paid more attention to better build an ecological civilization over the remote Three Rivers' Source region in the future.

Figure 12 shows the spatial distributions of the HCHO VCDs during the field campaign in July 2021. For the specific driving

routes (XD, DY or YX), the HCHO spatial distributions were similar on different days. Normally, the HCHO VCDs were lager at the starting points and ending points of the driving routes (if reaching to the ending points in the condition of SZA < 80°), which matched with the larger HCHO values in the morning and evening (Fig. 10). However, the HCHO levels were significantly different at the same location on different days. For example, the HCHO VCDs on the second circling journey (Fig. b1-b3) were obviously larger than those on the other three circling journeys, most probably due to higher surface

temperatures on the second circling journey (Fig. 12, S3). From the northeast to the southwest in the region of the mobile observation field experiment, the HCHO VCDs present a decreasing trend. These lower HCHO levels in the main area of Three Rivers' Source reflect the overall conditions of atmospheric HCHO background. The spatial distributions of HCHO column observed by the OMI satellite from 2009 to 2019 over the Tibetan Plateau also found that the regions with sparse population and less human activities were frequently affected by natural factors, such as air temperature and precipitation

(Zhang et al., 2021). The elevated HCHO VCDs around Maqin county of the XD driving route were partly related to anthropogenic HCHO emissions, such as biomass burning and fossil fuel combustion (Fig. a1, b1, c1, d1) (Zhang et al., 2021). Comparing the HCHO VCDs before and after Maduo county on the YX driving route, the former was larger than the later, corresponding to the jump of the HCHO diurnal variation before and after 13:00 BJT. Besides the differences in human activities, the spatial step changes in the HCHO VCDs were also partly connected with the decreasing altitudes on the YX

driving route (Fig. 2a).



## 4.4 Comparison with TROPOMI observations

The TROPOspheric Monitoring Instrument (TROPOMI) is the sole payload on the Copernicus Sentinel-5 Precursor (Sentinel-5P or S5P) satellite, which provides measurements of multiple atmospheric trace species including $NO_2$ and HCHO at high spatial and temporal resolutions (Veefkind et al., 2012). The S5P reference orbit is a near-polar sun-synchronous orbit with a mean Local Solar Time of 13:30 at Ascending Node. TROPOMI covers the wavelength ranges of ultraviolet-visible (270~495 nm), near infrared (675~775 nm), and shortwave infrared (2305~2385 nm) with a 108° Field-of-View in nadir view. TROPOMI achieves daily global coverage with a spatial resolution of $5.5 \times 3.5$ $km^2$ at nadir since the along-track pixel size reduction on August 6, 2019. The $NO_2$ retrieval consists of a three-step procedure: (1) The total $NO_2$ SCDs are retrieved from the Level-1b spectra measured by TROPOMI using the DOAS method; (2) The total $NO_2$ SCDs are separated into stratospheric SCDs and tropospheric SCDs on the basis of information coming from a data assimilation system; (3) The tropospheric $NO_2$ SCDs are converted into VCDs through a look-up table of tropospheric AMFs. The 1st and 3rd steps also apply to HCHO, but in addition, a bias of the HCHO SCDs needs to be corrected before the conversion of the HCHO SCDs to VCDs. In this study, we use the TROPOMI level-2 $NO_2$ and HCHO products (i.e. S5P_L2__NO2____HiR and S5P_L2__HCHO___HiR) downloaded from the NASA Goddard Earth Sciences Data and Information Services Center (GES-DISC) (ESA and KNMI, 2021; ESA and DLR, 2020). For comparison between the mobile MAX-DOAS and TROPOMI observations, their $NO_2$ and HCHO VCDs are gridded into 0.25 °×0.25 °cells (Fig. 13, 14).

Figure 13 shows the spatial distributions of the tropospheric gridded $NO_2$ VCDs from TROPOMI on each day of the field campaign. The spatial distributions of the tropospheric $NO_2$ VCDs are basically consistent on different days, i.e. higher values are found in the northeast and lower values in the southwest. Similar as for the mobile MAX-DOAS, the TROPOMI $NO_2$ VCDs are larger around Xining city than in the main area of Three Rivers' Source region. But the elevated trends of the tropospheric $NO_2$ VCDs around the counties, which are clearly observed by the mobile MAX-DOAS, are nearly not captured by TROPOMI. To validate the fine-scale spatial variability in tropospheric $NO_2$ VCDs, we made a linear regression analysis between both data sets (Fig. 15a). When using all tropospheric $NO_2$ VCDs at the same grid cell on the same day during the field campaign (referred to 'All' in Fig. 15a, corresponding to the white circles in Fig. 13), the consistency is good with a correlation coefficient of R=0.67 between the two data sets. However, the slope is much lower than unity indicating that TROPOMI systematically underestimates the $NO_2$ VCD over the polluted areas, in agreement with previous studies. Interestingly, there is almost no correlation of the two data sets, if we only use the tropospheric $NO_2$ VCDs within the 1.5 h time difference between mobile MAX-DOAS and TROPOMI at the same grid (referred to '$\Delta T_{1.5}$' in Fig. 15a, corresponding to the red pluses in Fig. 13). Comparing the situations of 'All' and '$\Delta T_{1.5}$', significant differences in the correlation are connected with the former including the larger $NO_2$ VCDs close to the cities, inferred by the locations of the grid cell in Fig. 13. For the '$\Delta T_{1.5}$' comparison, mostly the low background values are included. These results indicate the TROPOMI can distinguish the differences in tropospheric $NO_2$ VCDs between city and background atmosphere, but can't identify the fine-scale spatial variability in the tropospheric $NO_2$ VCDs in background atmosphere over the Tibetan Plateau. As a whole, in



contrast to routine TROPOMI validation based on site observations (Verhoelst et al., 2021), the mobile MAX-DOAS
observations can serve as a supplement to quantify the impact of the fine-scale $NO_2$ horizontal variability on satellite products.

In contrast to $NO_2$, the spatial distributions of the tropospheric gridded HCHO VCDs from TROPOMI are not uniform among different days of the field campaign (Fig. 14). The higher HCHO VCDs appear more in the second circling journey and the lower HCHO VCDs in the third and fourth circling journey, consistent with the aforementioned results derived from mobile MAX-DOAS. The HCHO levels around the city of Xining are also not significantly enhanced, even lower than those
in the main area of the Three Rivers' Source region on some days, such as 25 July 2021. We also perform a linear regression analysis of tropospheric HCHO VCDs derived from mobile MAX-DOAS and TROPOMI, respectively. Whether for 'All' (corresponding to the white circles in Fig. 14) situation or for '$\Delta T_{1.5}$' (corresponding to the red pluses in Fig. 14) situation, the correlation coefficients are the same (R=0.26 in Fig. 15b), indicating that there are no strong anthropogenic HCHO sources along the driving routes even in the city of Xining. The rather small correlation coefficient between the two data sets is also
related to the rather small variability of the HCHO VCDs and the relatively large noise in the TROPOMI satellite product, which prevents to monitor the fine-scale spatial variability in tropospheric HCHO VCDs in background atmosphere over the Tibetan Plateau. Comparing the '$\Delta T_{1.5}$' situation between $NO_2$ and HCHO, the correlation of the tropospheric HCHO VCDs is higher than that of $NO_2$, which is probably related to the stronger HCHO daily variations in the background atmosphere influenced by natural factors, such as air temperature and precipitation (Zhang et al., 2021). Similar to the validations of
TROPOMI at remote sites by ground-based solar-absorption Fourier-transform infrared (FTIR) measurements (Vigouroux et al., 2020), an overestimation of the true HCHO VCD by TROPOMI is also found during the field campaign, with significantly larger relative differences of 104% and 87% for 'All' and '$\Delta T_{1.5}$' on average, respectively (Fig. 15b). Therefore, although TROPOMI significantly improves the precision of the HCHO observations at short temporal scales and for low HCHO columns (De Smedt et al., 2021), it is still very difficult for satellite instruments to detect the fine-scale spatial and temporal
variations of HCHO over the Tibetan Plateau.

## 5 Summary and conclusions

In this study we performed mobile MAX-DOAS measurements over the Tibetan Plateau in summer (18–30 July) 2021 for the first time. We analysed spectra of scattered sun light collected in the Three Rivers' Source region over the Tibetan Plateau, and obtained the data sets of tropospheric $NO_2$ and HCHO VCDs in the background atmosphere with high spatio-temporal
resolution; We further investigated the abundances and spatio-temporal variations of the tropospheric $NO_2$ and HCHO VCDs, and validated the TROPOMI satellite products during the field campaign.

We tested the influences of different Fraunhofer reference spectra (FRSs) and different spectral intervals on the spectral retrieval, and found that the fitting residuals are smaller when using the sequential FRSs in the $NO_2$ visible wavelength region for mobile MAX-DOAS measurements in the background atmosphere over mountain terrain. After investigating the optimal
filters to eliminate the "bad measurements" caused by sunlight shelters and vehicle's vibration and bumpiness, the $NO_2$ and





HCHO DSCDs were retained with the conditions of (1) RMS < 0.005, (2) offset (constant) between ± 0.03, and (3) SZA < 80°. The qualified $NO_2$ and HCHO DSCDs were converted to the corresponding VCDs based on the air mass factor (AMF) estimated by the geometric approximation method. Through comparing the resulting $NO_2$ and HCHO VCDs at three different elevation angles (15°, 20°, 30°), the $VCD_{15°}$ were further filtered and kept as the final data sets of tropospheric $NO_2$ and HCHO

VCDs when absolute and relative VCD differences (ΔVCD) between 15° and 20° are < $10^{15}$ molec $cm^{-2}$ or <5% for $NO_2$ and < $2\times10^{15}$ molec $cm^{-2}$ or <5% for HCHO, respectively.

The background levels of tropospheric $NO_2$ and HCHO VCDs, estimated by the maximum frequency method, were 0.40 $\times10^{15}$ molec $cm^{-2}$ for $NO_2$ and 2.27 $\times10^{15}$ molec $cm^{-2}$ for HCHO in July 2021 over the Three Rivers' Source region. We also determined the dependence of the tropospheric $NO_2$ and HCHO VCDs on altitude, which generally presents a decreasing trend

with the increasing altitude. This characteristic for natural background atmosphere is probably mainly related to the lower air density at higher altitude. However, different from the nearly constant decreasing rate of HCHO VCDs with increasing altitude, the differences of decreasing rate above and below the 2750 m altitude for $NO_2$ VCDs are significant, which is highly connected with different contributions of anthropogenic sources and natural sources for $NO_2$ and HCHO.

The $NO_2$ daily means were always larger than the corresponding medians on each day, but the daily means and medians for

HCHO were very close to each other. The day-to-day variations of the $NO_2$ VCDs between different circling journeys were similar, i.e. similar geographical distributions of the $NO_2$ VCDs were observed for each circling journey. However, the daily HCHO VCDs varied from the minimum mean of 1.15 $\times10^{15}$ molec $cm^{-2}$ on 27 July 2021 to the maximum mean of 4.63 $\times10^{15}$ molec $cm^{-2}$ on 21 July 2021. The obvious differences of the daily HCHO VCDs between different circling journeys were highly related to the air temperature, which affects the levels of VOCs emitted by natural vegetation. The daytime diurnal

cycles of $NO_2$ VCDs presented higher values in the morning and evening, and the amplitudes of the $NO_2$ diurnal variation changed with the driving routes. The simulations of stratospheric $NO_2$ absorption cannot explain the observed diurnal cycle of the tropospheric $NO_2$ VCD. Besides the enhanced $NO_2$ photolysis around noon, the enhanced $NO_2$ VCDs in the morning and evening were primarily caused by enhanced pollution when the mobile observation vehicle was located in or close to the cities or county town.

With respect to the spatial distributions, the tropospheric $NO_2$ VCDs over the main area of Three Rivers' Source were relatively uniform with very low levels (<$1\times10^{15}$ molec $cm^{-2}$), but they were usually elevated in cities or counties, around the Qinghai Lake, even occasionally at the highway junction and the tunnel exit, where there were enhanced transport emissions. Overall, the HCHO VCDs presented a decreasing trend from the northeast to the southwest in the region of the field experiment. The HCHO VCDs were elevated at the starting points and ending points of the driving routes, corresponding to larger HCHO

VCDs in the morning and evening. The levels of the HCHO VCDs were variable on different days at the same location, implying that natural factors, such as air temperature, significantly influenced the atmospheric HCHO photochemical formation.



TROPOMI $NO_2$ clearly presents the obvious influences of anthropogenic sources on enhanced $NO_2$ VCDs around Xining city. In contrast, the stronger influences of natural factors on HCHO lead to larger daily variation of HCHO, which causes inconsistent and variable spatial distributions of TROPOMI HCHO VCDs on different days but also a higher correlation between mobile MAX-DOAS and TROPOMI than $NO_2$ for the background atmosphere. In addition, through comparing $NO_2$ and HCHO VCDs between mobile MAX-DOAS and TROPOMI, we found that TROPOMI can distinguish the differences in tropospheric $NO_2$ VCDs between city and background atmosphere, but cannot identify the fine-scale spatial variability in tropospheric $NO_2$ and HCHO VCDs in the background atmosphere over the Tibetan Plateau.

As a whole, we obtained valuable data sets and information of the spatio-temporal variation of $NO_2$ and HCHO over the Tibetan Plateau, which have the great potential in investigating the evolution of the atmospheric composition with high spatio-temporal resolution in the background atmosphere at high altitude, validating and improving the satellite products over mountain terrain, and evaluating atmospheric chemistry model over the Tibetan Plateau.



**Code and data availability.** The filtered final NO₂ and HCHO VCDs for the field campaign by mobile MAX-DOAS
observations on 18-30 July 2021 over the Three Rivers' Source region of the Tibetan Plateau in China are available upon request.

**Supplement.** The supplement related to this article is available online.

**Author contributions.** X.H. Cheng and X.D. Xu designed the field experiment. S.Y. Cheng and J.Z. Ma set up the mobile MAX-DOAS measurement platform under discussions with X.H. Cheng, J.G. Lv, S. Dörner, S. Donner, and T. Wagner. W.Q.
Zhang, G. Bai, B. Chen, and S.Y. Ma contributed to the field measurements. S.Y. Cheng performed the spectra retrieval and data analysis with contributions from T. Wagner, S. Dörner, S. Donner, and J.Z. Ma. S.Y. Cheng, J.Z. Ma, and T. Wagner prepared the manuscript with consent by all co-authors.

**Competing interests.** The authors declare that they have no conflict of interest.

**Acknowledgements.** We thank the staff at the Qinghai Meteorological Administration for supporting the measurements. We
thank BIRA-IASB for QDOAS spectral analysis software. We also thank ESA, KNMI, DLR and NASA for the TROPOMI satellite products.

**Financial support.** This research is supported by grants from the Fundamental Research Funds for Central Public-interest Scientific Institution from Chinese Academy of Meteorological Sciences (No. 2021Z013), the National Natural Science Foundation of China (No. 41875146), and the Fund of State Key Laboratory of Applied Optics (No. SKLAO2021001A02).





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



**Table 1.** Observation periods and routes of the mobile MAX-DOAS field experiment over the Three Rivers' Source region of the Tibetan Plateau in July 2021.

| Cycles | Xining to Dari (XD) | Dari to Yushu (DY) | Yushu to Xining (YX) |
|---|---|---|---|
| 1 | 2021-07-18 9:00~22:49BJT[a] | 2021-07-19 9:05~17:40BJT | 2021-07-20 8:17~21:48BJT |
| 2 | 2021-07-21 8:09~21:40BJT | 2021-07-22 8:20~16:07BJT | 2021-07-23 8:18~21:38BJT |
| 3 | 2021-07-25 8:29~20:08BJT | 2021-07-26 8:08~15:20BJT | 2021-07-27 8:18~21:48BJT |
| 4 | 2021-07-28 8:27~18:56BJT | 2021-07-29 9:00~16:00BJT | 2021-07-30 8:21~22:35BJT |

[a] BJT denotes the Beijing time, corresponding to Universal Time Coordinated (UTC) + 8 h.

**Table 2.** Fit settings for the $NO_2$ and HCHO spectral analyses.

| Parameters | Setting for $NO_2$ | Setting for HCHO |
|---|---|---|
| Fraunhofer reference spectrum | sequential spectra | sequential spectra |
| fitting interval (nm) | 400~434 | 324~359 |
| DOAS polynomial | degree: 5 | |
| intensity offset | degree: 2 (constant and order 1) | |
| shift and stretch | spectrum | |
| Ring spectra | original and wavelength-dependent Ring spectra | |
| $NO_2$ cross section | Vandaele et al. (1998), 294 K, $I_o$ correction ($10^{17}$ molec•cm$^{-2}$) | |
| $H_2O$ cross section | Polyansky et al. (2018), 293 K | / |
| $O_3$ cross section | Serdyuchenko et al. (2014), 223 K, $I_o$ correction ($10^{20}$ molec•cm$^{-2}$) | Serdyuchenko et al. (2014), 223 K, 243 K, $I_o$ correction ($10^{20}$ molec•cm$^{-2}$) |
| $O_4$ cross section | Thalman and Volkamer (2013), 293 K | Thalman and Volkamer (2013), 293 K |
| HCHO cross section | / | Meller and Moortgat (2000), 298 K |


**Table 3.** Correlation for the $NO_2$ and HCHO VCDs between the three elevation angles ($15°$, $20°$, $30°$).

| Correlation Coefficient | $NO_2$ | HCHO |
|---|---|---|
| $R_{15°, 20°}$ | 0.95 | 0.80 |
| $R_{15°, 30°}$ | 0.91 | 0.66 |
| $R_{20°, 30°}$ | 0.94 | 0.73 |





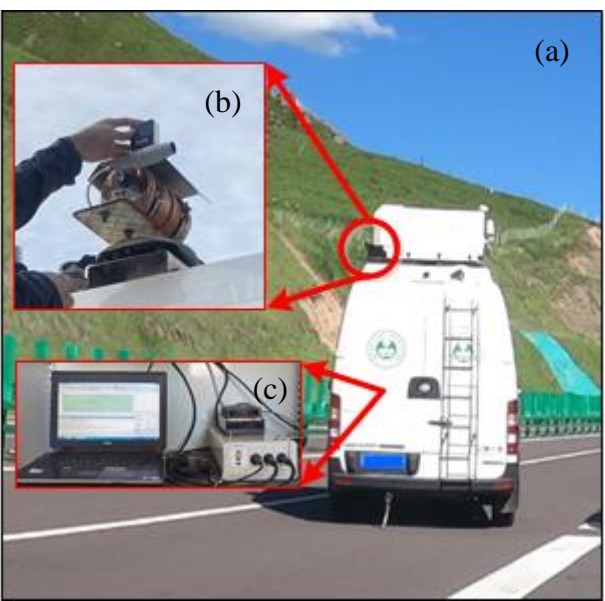

**Figure 1. (a)** Mobile observation vehicle of atmospheric composition and meteorological parameters. Two parts of the Tube MAX-DOAS instrument are installed **(b)** on the rear of the vehicle's roof and **(c)** inside the vehicle, respectively.

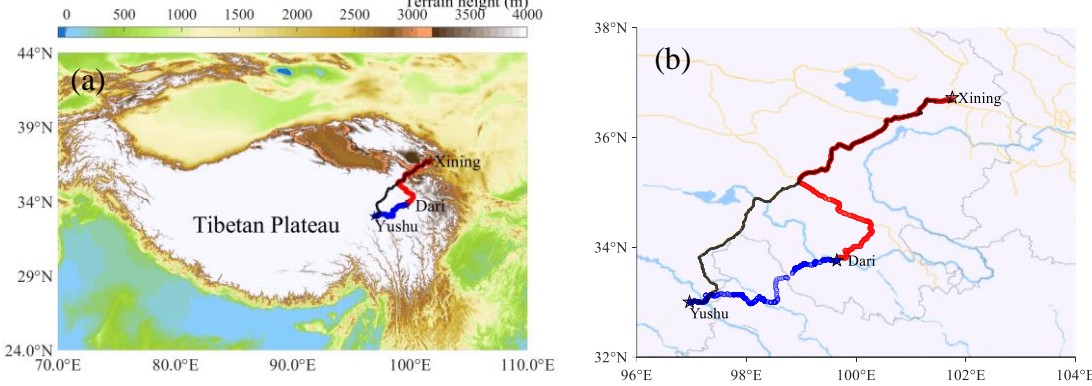

**Figure 2.** Driving routes (red, blue and black lines) of the mobile observation vehicle. The driving routes are added to **(a)** the terrain height map over the Tibetan Plateau and **(b)** the street map (https://map.baidu.com/, last access: 16 June 2022) in the experiment region as an overlay, respectively. Light blue lines and areas in figure (b) indicate rivers and lakes.





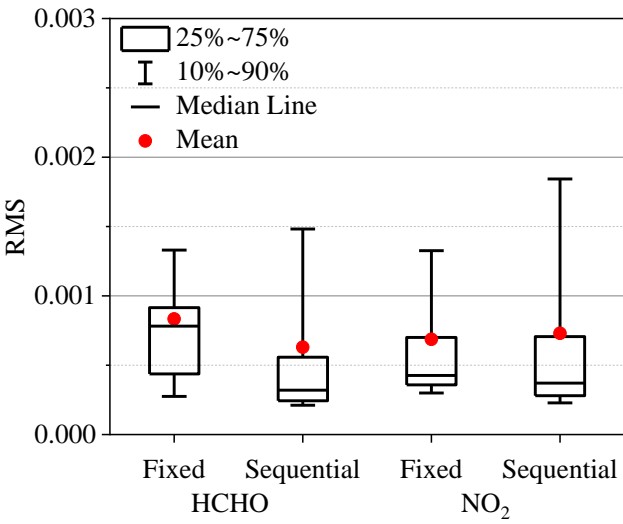


**Figure 3.** Statistics of the root mean square (RMS) of the $NO_2$ and HCHO spectral fitting residuals using a sequential FRS or fixed FRS (for RMS<0.005 and SZA<80°) during the field campaign. Lower (upper) error bars and boxes are the 10th (90th), 25th (75th) percentiles, respectively. Lines inside the boxes and dots denote the medians and the mean values.

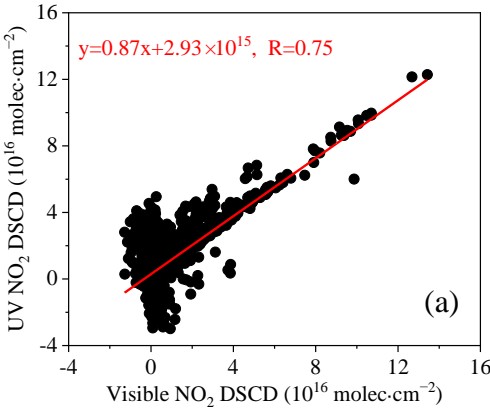
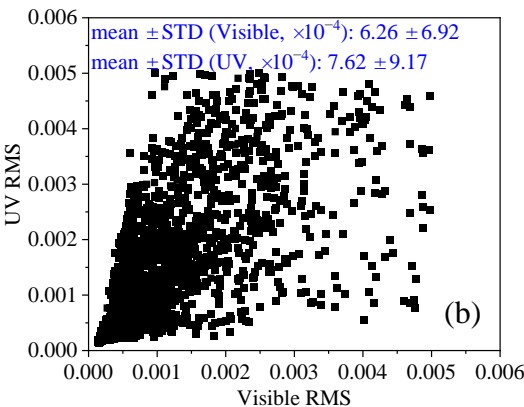

**Figure 4.** Comparison of $NO_2$ spectral fitting results using the visible and UV wavelength intervals (for RMS<0.005 and SZA<80°) during the field campaign. **(a)** Linear fit of corresponding $NO_2$ DSCDs between visible and UV spectral intervals. **(b)** Corresponding $NO_2$ RMS between visible and UV spectral intervals. The red lines denote the results of the regression analyses and the corresponding equations and correlation coefficients are displayed in the figure (a). The numbers in figure (b) indicate the mean ±standard deviation (STD) in the visible and UV spectral intervals.





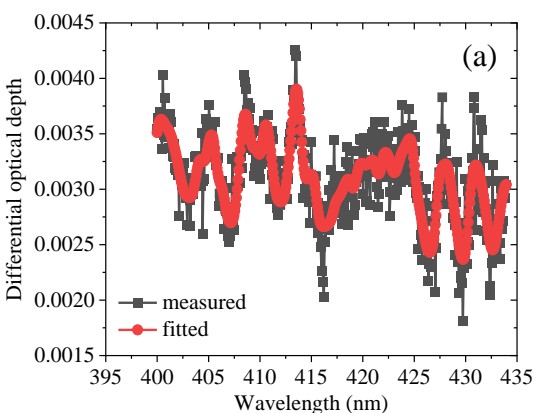 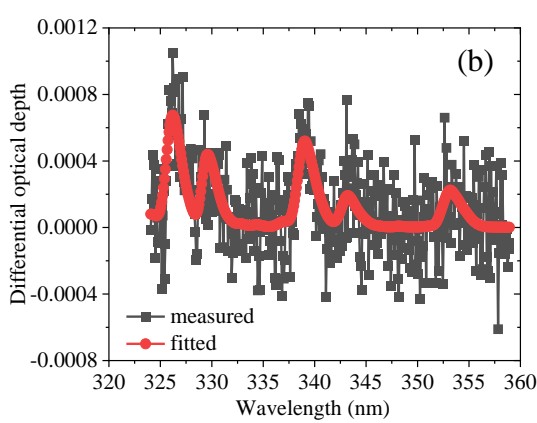

**Figure 5.** Examples of DOAS spectral analyses for **(a)** NO₂ and **(b)** HCHO. Black curves with squares and red curves with dots indicate the measured and fitted differential optical depth, respectively. The NO₂ and HCHO DSCDs are $5.27 \times 10^{15}$ molec cm$^{-2}$ and $9.36 \times 10^{15}$ molec cm$^{-2}$, respectively. The RMSs of the spectral fitting residuals between measured and fitted spectra are $2.17 \times 10^{-4}$ for NO₂ and $2.09 \times 10^{-4}$ for HCHO, respectively.

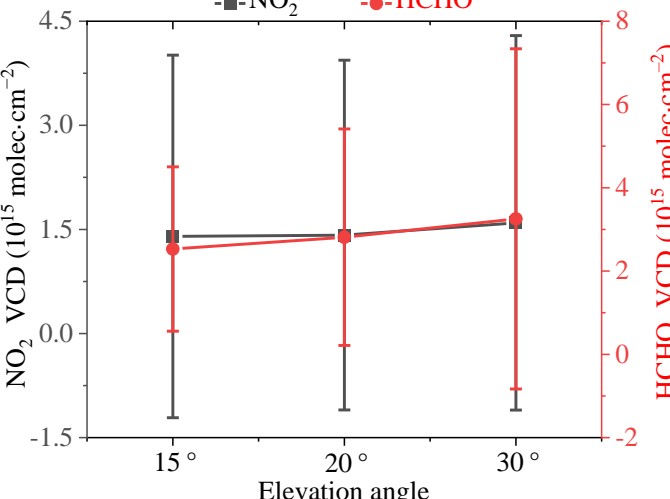

**Figure 6.** Mean NO₂ (line with squares) and HCHO (line with dots) VCDs calculated by the geometric approximation method for different elevation angles during the effective observation period. The error bars indicate the standard deviations of NO₂ and HCHO VCDs for different elevation angles.





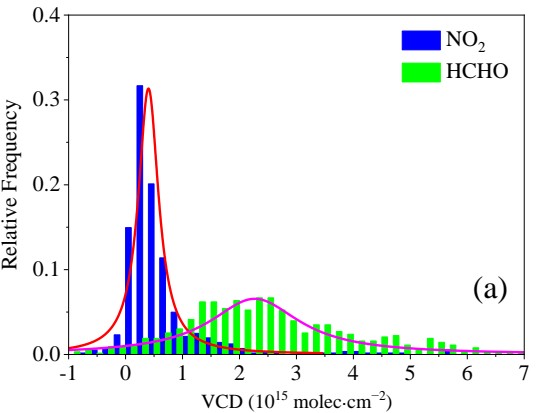
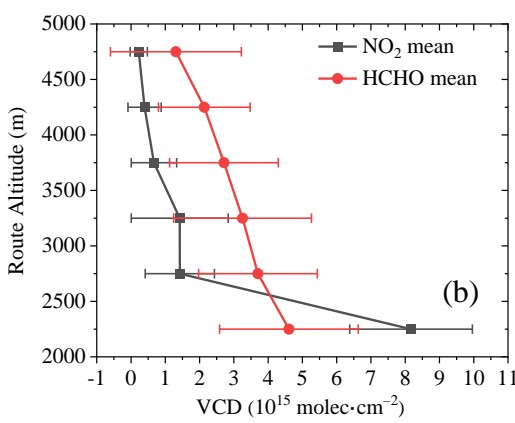

**Figure 7.** Overall characteristics of $NO_2$ and HCHO VCDs during the field campaign. **(a)** Frequency distributions of $NO_2$ (blue column) and HCHO (green column) VCDs as well as their Lorentz distribution curves for $NO_2$ (red curve) and HCHO (magenta curve), respectively. **(b)** Dependence of the $NO_2$ and HCHO VCDs on altitude from 2000 m to 5000 m at vertical intervals of 500 m. The black (red) lines, symbols and error bars denote the means and standard deviations of the $NO_2$ (HCHO) VCDs for each altitude range.

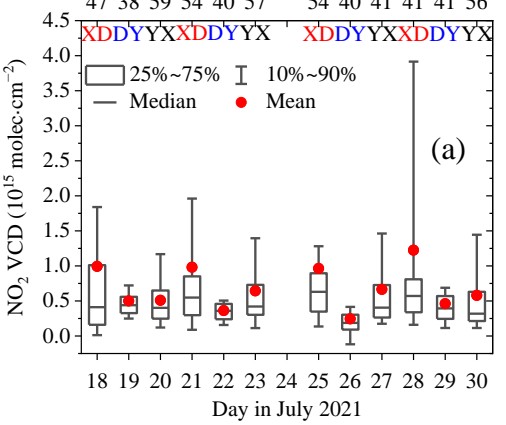
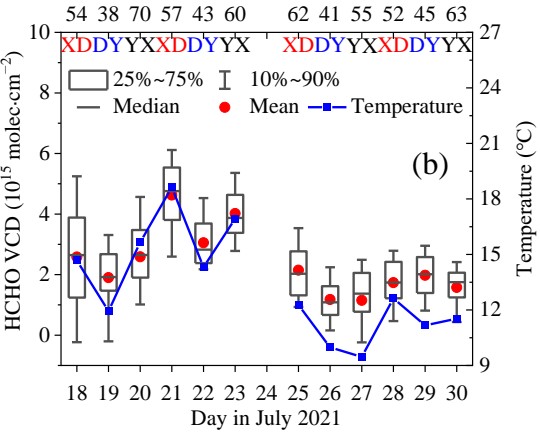

**Figure 8.** Day-to-day variations of the daily averaged **(a)** $NO_2$ and **(b)** HCHO VCDs over the mobile observation routes (XD, DY, YX). Lower (upper) error bars and boxes are the 10th (90th), 25th (75th) percentiles of the data. Lines inside the boxes and red dots denote the medians and the mean values, respectively. The integrated sampling numbers for specific day are labelled at the top axis. The blue curves with squares in Figure (b) denote the daily air temperature at 2 m above the land surface.



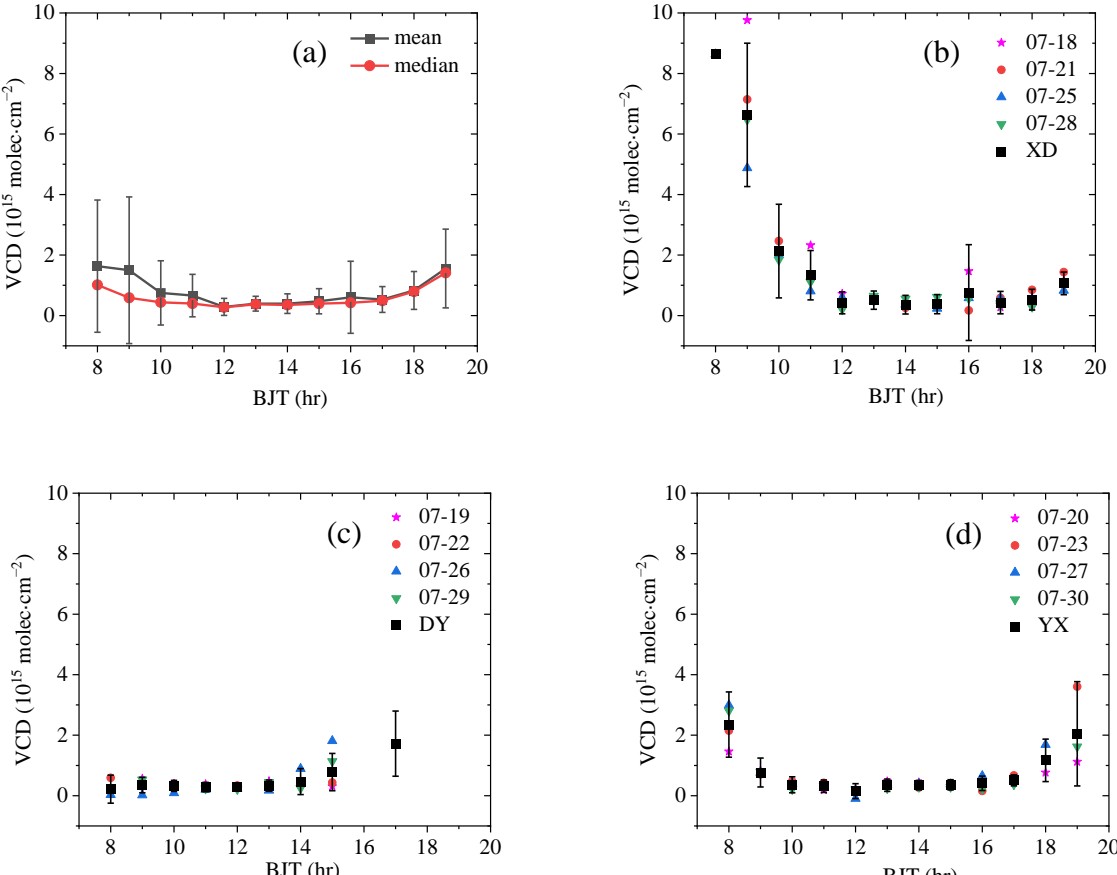

760

**Figure 9.** Diurnal variations of the NO$_2$ VCDs over the mobile observation routes. **(a)** Diurnal variations of the overall means (black curve with squares), medians (red curves with dots), and standard deviations (error bars) of the NO$_2$ VCDs. **(b)** Diurnal variations of the mean NO$_2$ VCDs on selected days (18/21/25/28 July 2021) as well as the means and standard deviations of the NO$_2$ VCDs on the XD driving route. **(c, d)** Same as (b), but for the DY and YX driving routes during the field campaign.





765

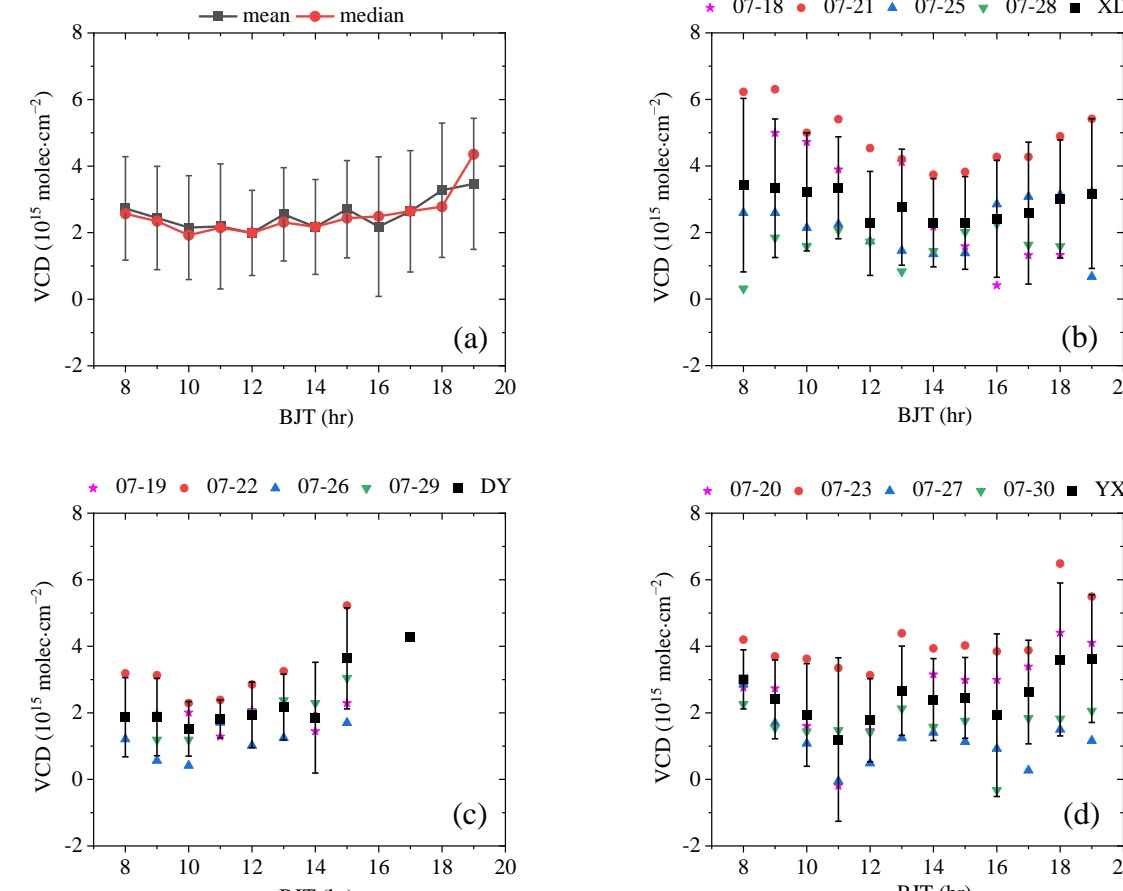

**Figure 10.** Same as figure 9, but for HCHO.





**Figure 11.** Spatial distributions of gridded NO₂ VCDs with 0.25 °×0.25 ° resolution. The observed NO₂ VCDs in each spatial grid cell are
770    averaged for three segments (**1, 2, 3**) of four circling journeys (**a, b, c, d**). The main cities and counties on the driving routes are marked by
the black stars. On the background map, the light blue lines and areas represent rivers and lakes (such as, Qinghai Lake), the yellow lines
denote the roads, and the grey lines indicate the administrative boundaries.







**Figure 12.** Same as figure 11, but for HCHO.





**Figure 13.** Spatial distributions of the tropospheric NO₂ VCDs observed by TROPOMI on each day of the field campaign. The TROPOMI S5P_L2__NO2____HiR product has been gridded to 0.25 °×0.25 ° cells. The main cities and counties on the driving routes of the field campaign are marked by the black stars. The black curves indicate the administrative boundaries. The white circles and red plus symbols show the grid cell where the data of both TROPOMI and MAX-DOAS are available on the same day or within a 1.5 h time difference, respectively.





**Figure 14.** Same as figure 13, but for HCHO.



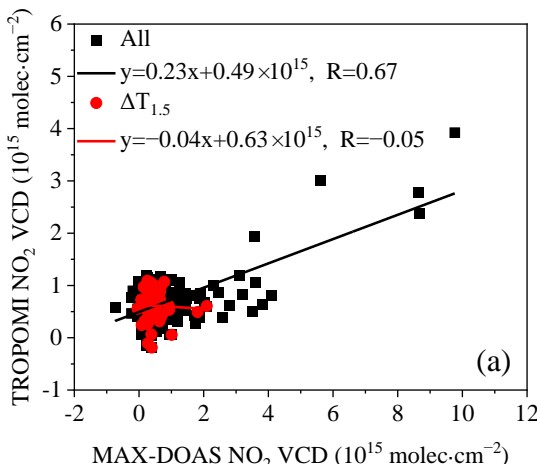
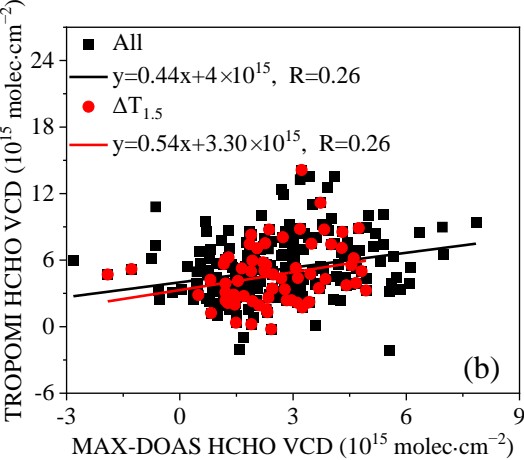

**Figure 15.** Linear fit between the tropospheric **(a)** NO$_2$ and **(b)** HCHO VCDs measured by the mobile MAX-DOAS and TROPOMI. The black squares and red dots represent the available VCDs of both data sets at the same grid cell on the same day or within a 1.5 h time difference, respectively. The black (red) lines denote the results of the regression analyses and the corresponding equations and correlation coefficients are displayed in the figures.