# Peer review of "Mobile MAX-DOAS observations of tropospheric NO2 and HCHO during summer over the Three Rivers' Source region in China"

_Atmospheric Chemistry and Physics, 2022_

## Author Comment (AC1)

Referee comments are in black. Author responses are in blue.

This manuscript describes mobile MAX-DOAS measurements recorded on drives around the Tibetan Plateau. The measurements are used to quantify the column of NO₂ and HCHO along circular drive paths on the Plateau. The measurements are compared to TROPOMI satellite products. The manuscript represents a valuable contribution to the literature and reports ground truth around this relatively remote and high-altitude region. The manuscript uses the geometric method to retrieve tropospheric vertical column densities from differential slant column densities measured on the vehicle. This method, while lacking the refinement of one using radiative transfer calculations, is reasonable for this purpose. However there are some details on the method that should be discussed further and it would be valuable for other groups to understand how to optimize the measurements to get the most measurements from such studies. Concerns on the method are described below, followed by specific comments. If the concerns regarding the method can be addressed, this manuscript would be acceptable for publication in ACP.

Reply: First of all, we appreciate the reviewer's positive comments on our manuscript. In response to the reviewer's comments and suggestions, we have made relevant revisions to the manuscript. Listed below are our responses and the corresponding changes made to the manuscript according to the suggestions given by the reviewer. Note that the Sections 4.2 (Temporal variation) and 4.3 (Spatial distribution) in the original manuscript were reorganized into Section 4.2 (Spatio-temporal variation) in the revised manuscript. This change is not marked up using revision track in order to keep the manuscript clear to read.

Were the elevation angles recorded relative to the mobile vehicle or to gravity? If they were gravitationally referenced, was a gyroscope used? If they are relative to the road, how is the local horizon taken into consideration? I think lines 260-265 indicate that the angles are with respect to the vehicle, and that is a reason to use

higher elevation angles, but the text is not very clear to this regard. Can the authors be more clear about how the view geometry is defined?

Reply: The elevation angles were recorded relative to the mobile vehicle. We didn't use a gyroscope. During our deployment strategies, we designed a partial system recording the attitude angles of the mobile vehicle to correct the elevation angle of MAX-DOAS measurements. However, it did not work well and couldn't be used for this study. To reduce the influences of local non-horizontal road on the un-corrected elevation angle, we used the DSCDs at larger elevation angles. Also, the VCDs were further filtered based on the absolute difference and the relative difference of VCDs between $15\,^{\circ}$ and $20\,^{\circ}$. The definition of elevation angle has been added to the text in Section 2.1 of the revised manuscript.

As is pointed out by the authors, the higher altitude and lower aerosol extinction conditions of a sparsely populated plateau make the geometric approach more tenable. However, clouds or aerosol-particle-rich pollution plumes (which may accompany NO2) may affect this assumption. Therefore, it seems reasonable to check this assumption by using O4 observations. The authors could calculate the O4 VCD above the vehicle from the altitude (which they know) and pressures at meteorological stations (or better soundings if available). They can then use the mobile-measured geometrically calculated O4 VCD_15 to compare to the meteorologically calculated one. Due to radiative transfer effects, particularly the relative azimuth angle to the sun, these quantities won't be perfect, but times when there is a large amount of aerosol or clouds or the view azimuth happened to be close to the sun, one could tell that the NO2 and HCHO data are being affected by these confounding effects. This check should help to assure that the geometric method is working for NO2 and HCHO.

Reply: Many thanks for your suggestions. The procedures of $O_4$ VCDs derived from MAX-DOAS measurements and calculations are as the following.

(1) The $O_4$ DSCDs were retrieved from MAX-DOAS spectra in the wavelength interval of 351-390 nm using a sequential FRS and are then filtered by the conditions of SZA $<80\,^{\circ}$, RMS $< 0.005$, offset (constant) between $\pm 0.03$. The $O_4$ VCDs obtained by the geometric approximation method at $15\,^{\circ}$ elevation angle were further filtered by the differences of the $O_4$ VCDs between $15\,^{\circ}$ and $20\,^{\circ}$. The $O_4$ VCDs were kept if the

absolute difference of VCDs between $15°$ and $20°$ is $< 1 \times 10^{42}$ molec$^2$ cm$^{-5}$ or the relative difference is <10%.

(2) During the procedure of the $O_4$ VCD calculation, we used the air temperature and pressure profiles of hourly ERA5 with $0.25°\times0.25°$ grid above the altitude of the driving route. Firstly, we extracted the profiles of temperature and pressure matched with each measurement at the same grid cell and the same hour. Then we calculated the $O_2$ concentrations from the surface to 30 km at each altitude with a vertical interval of 50m. The $O_4$ concentrations were assumed as the square of the $O_2$ concentrations at each vertical grid cell and integrated as $O_4$ VCDs from the surface to 30 km (Wagner et al., 2019).

The results of $O_4$ VCDs derived from the MAX-DOAS measurements and calculated from ERA5 data are shown in the figure below. The main finding is that the measured $O_4$ VCDs are systematically lower than the calculated ones. Part of the underestimation is probably related to clouds, but a strong underestimation is also found for measurements for clear skies.

[Figure]

Hence we further explored the applicability of the geometric approximation method by radiative transfer simulations. According to the AODs from the AERONET website (https://aeronet.gsfc.nasa.gov, last access: 2 December 2022) at three sites (Mt_WLG, NAM_CO, QOMS_CAS) over the Tibetan Plateau, we estimate the AODs

around 0.1 during our field campaign. But similar results are also found for AODs of 0.05 and 0.2. The simulation scheme is as the following.

| Parameters | $O_4$ | $NO_2$ | HCHO |
|---|---|---|---|
| Wavelength (nm) | 340 | 440 | 340 |
| Layer height (km) | US standard atmosphere | 0-1; 0-2 | 0-1; 0-2 |
| Aerosol height (km) | 0-1 | same as trace gases | |
| AOD | 0; 0.05; 0.1; 0.2 | 0; 0.05; 0.1; 0.2 | 0; 0.05; 0.1; 0.2 |
| SZA (°) | 20, 40, 60, 70, 80 | | |
| RAA (°) | 10, 30, 60, 90, 180 | | |
| Elevation angle (°) | 15 | | |
| Terrain height (km) | 2, 3, 4, 5 | | |

The VCD ratios of the RTM simulations and the geometric approximation for 15 ° elevation angle under the condition of AOD=0.1 can be obtained for $O_4$, $NO_2$, and HCHO, respectively. The DAMF ratios' means and standard deviations for all geometries (blue symbols) and RAA=10 °, SZA=60 ° and RAA=10 °, SZA=70 ° excluded (red symbols, for these rare measurement scenarios the strongest errors occur) are shown below. The main findings are: (1) The typical errors of the geometric approximation are <20% for $NO_2$ and HCHO; (2) The errors of the geometric approximation are much larger for $O_4$ with a systematic underestimation between about 40% and 60%, which are in overall agreement with the comparison of the measured and calculated $O_4$ VCDs above; (3) The large underestimation of the $O_4$ VCDs indicates that $O_4$ can not be used for the test if the geometric approximation is justified or not for an individual measurement of $NO_2$ and HCHO.

[Figure]

[Figure]

[Figure]

**Figure S1** in the revised supplement

**Reference:**

Wagner, T., Beirle, S., Benavent, N., Bösch, T., Chan, K. L., Donner, S., Dörner, S., Fayt, C., Frieß, U., García-Nieto, D., Gielen, C., González-Bartolome, D., Gomez, L., Hendrick, F., Henzing, B., Jin, J. L., Lampel, J., Ma, J., Mies, K., Navarro, M., Peters, E., Pinardi, G., Puentedura, O., Puķīte, J., Remmers, J., Richter, A., Saiz-Lopez, A., Shaiganfar, R., Sihler, H., Van Roozendael, M., Wang, Y., and Yela, M.: Is a scaling factor required to obtain closure between measured and modelled atmospheric $O_4$ absorptions? An assessment of uncertainties of measurements and radiative transfer

simulations for 2 selected days during the MAD-CAT campaign, Atmospheric Measurement Techniques, 12, 2745-2817, 10.5194/amt-12-2745-2019, 2019.

It seems like data were recorded at 7 elevation angles, but only four of them were used, and then the 15 degree angle was selected, so only two (15 and 90) were used for the final determination of tropospheric VCD. Therefore, the scan pattern seems inefficient. The authors should discuss good practices for mobile DOAS deriving from this experience. It seems like the upper elevation angles are useful to tell that 15 degrees is not biased compared to other angles, but the lower elevation angles are affected both by road tilt (if the geometry is based upon the vehicle -- see above) and obstructions (e.g. buildings, canyon walls, etc.). Can the authors discuss this issue and give advice for future studies?

Reply: Many thanks for your good comments and suggestions. Because this is the first practice of mobile MAX-DOAS observations over the Tibetan Plateau, we didn't know which elevation angle was the best for measuring the tropospheric VCDs of trace gases in the background atmosphere over mountain terrain before this campaign. Therefore, we made the telescope scanning at 7 elevation angles. In future studies on observing tropospheric $NO_2$ and HCHO VCDs by mobile DOAS, we suggest to measure at $15°, 20°, 90°$ elevation angles. There are at least two reasons: (1) The larger elevation angles were less influenced by the road tilt and obstructions; (2) The measurements at $15°$ and $20°$ elevation angles have an enhanced sensitivity to tropospheric trace gases (increase of sensitivity compared to $90°$ elevation is about a factor 3.8 and 2.9, respectively). The increased sensitivity is especially important for measurements of the rather low trace gas concentrations in the background atmosphere. We have added the suggestion to the text at the end of Section 3.2 in revised manuscript from our experiences in this study.

The writing of this manuscript is readable, but in places it could be condensed regarding details that don't seem relevant to the study. For instance, description of the study region seems to include details not really related to the purpose of the study. In places some phrases may also need minor English language editing to read more clearly.

Reply: Many thanks for your kind suggestions. The paragraph about the study

region has been refined. We also improved the English language in the revised manuscript.

**Specific comments:**

Units -- ACP uses SI units, which indicate that ppb and ppt are language dependent, so they prefer mixing ratios in nmol mol^-1 or pmol mol^-1.

Reply: Agreed. We have checked the units in this paper and revisions have been made.

Line 117: Maybe a transition here to say that although there are challenges, it is useful for reasons...

Reply: Agreed. The description has been modified as "Although there are challenges in measuring $NO_2$ and HCHO concentrations by mobile MAX-DOAS over the Tibetan Plateau, they are useful for studies on the spatio-temporal evolution of the atmospheric composition in the background atmosphere, validation and improvement of satellite products over mountain terrain, and evaluation of the simulation results of atmospheric chemistry models over the Tibetan Plateau.".

Line 150: Were the angles with respect to gravity or with respect to the mobile platform? How where they corrected to be with respect to gravity?

Reply: The angles were with respect to the mobile platform. We use the uncorrected elevation angles in this study. Originally we planned to use the platform attitude angle to correct the elevation angles. It is a pity that the partial system of the attitude angles of the mobile vehicle did not work well during the field campaign. Nevertheless, we estimated the uncertainties for measurements on tilted roads and found them very small (~1%) for the average of several measurements (for more details see below).

Line 152: What company manufactured the spectrometer?

Reply:     AVANTES.     This     information     and     model     number
(AvaSpec-ULS2048x64-USB2) have been added to the revised manuscript.

Lines 162 to 174: Some of this repeats information in the introduction, and some are a bit challenging to read (e.g. what does "four indistinct seasons" mean?).  I'd suggest making this section more directly relevant to the mobile campaigns.

Reply: Many thanks for your kind suggestions. This paragraph has been refined.

Line 202: I found this sentence confusing.  You could possibly reword or add the word "respectively" after "... can be neglected or cancels out".  I think you mean that if a species has no stratospheric part, you can neglect the SCD_stra, or in the other case, if a species has a stratospheric part and there is no light scattering in the stratosphere (thus the light path in the stratosphere is the same independent of alpha) that SCD_stra appears in both SCDs and will then cancel out.

Reply: Per your suggestion, we have added the word "respectively". You understood this sentence correctly.

Line 209: I presume the interpolation is in time at which the off-zenith spectrum occurs weighting the two neighboring zenith spectra.  Can you clarify?

Reply: Yes, the "sequential FRS" are defined as the time interpolated spectra between two zenith spectra measured before and after the measurement time of the current off-zenith elevation angle. We have amended the description in the revised manuscript.

Line 266: I think that the authors should estimate the effect of the elevation angle error.  Presuming that the view is relative to the car, one could use an estimate of road grade angle to calculate the magnitude of this error.  In the US, interstate highways are allowed to be up to 6% grade (angle = arctan(0.06) = 3.4°).  It would be good to quantify the magnitude of this error, and while I expect it to be small compared to others, the authors should show that it "can be neglected".

Reply: Many thanks for your kind suggestion. we estimate the error of the elevation angle to be about 2.3°, based on the median of the mobile platform attitude angle

during the effective MAX-DOAS measurement period. The corresponding error of an individual measurement will be up to about 21%. However, it should be noted that on average the positive and negative deviations of the elevation angle will almost cancel each other. Thus the errors of individual measurements will be usually much smaller (except for measurements on continuous strong slopes). For averages of several measurements the errors of the elevation angles lead to much smaller VCD errors with a magnitude smaller than 1% when using geometric approximation method (equation 6):

$\alpha = 15° - 2.3° = 12.7°$, VCD=0.2818×DSCD;

$\alpha = 15°$, VCD=0.3492×DSCD;

$\alpha = 15° + 2.3° = 17.7°$, VCD=0.4232×DSCD;

[(0.2818×DSCD+0.4232×DSCD)÷2-0.3492DSCD] ÷(0.3492×DSCD)×100 =1%

Line 270: I think that this implies that "the geometric approximation method is self consistent", but not that it "has high accuracy". The test done by the authors is only a test of how consistent their data at one elevation angle is compared to another of their elevation angles. If there were aerosol light extinction that reduced pathlengths on each view, the results would still be correlated, but would be affected and not be accurate.

   Reply: Agreed. The description has been amended as "the geometric approximation method is self-consistent".

Line 320: I think the wording "This implies that..." is a bit too strong. The HCHO data are consistent with increasing temperature leading to more BVOC emissions, but they could also be affected by the temperature of the photochemical sources and sinks of HCHO.

   Reply: Many thanks for your suggestion. The description of "This implies that" has been amended to "Probably" in the revised manuscript.

Around line 341: Could the U-shape for NO2 also be affected by the city at the start and end of each daily journey?

Reply: Yes. The U-shape of $NO_2$ VCD diurnal variation was affected by several factors. From our findings we conclude that the $NO_2$ diurnal variations were primarily caused by enhanced pollution in the morning and evening when the mobile observation vehicle was located in or close to the cities or county town. An additional effect on the diurnal variation is probably caused by the enhanced $NO_2$ photolysis around noon.

Line 394: Are these figure numbers right? I'm not sure how I can tell about the telescope direction from these maps. Possibly some better annotation on the maps (e.g. an arrow or special marker) would help. I'm not sure what "vehicle flowrate was less" means.

Reply: We checked the figure numbers again and they are correct. According to the explanations of the driving routes in Table 1, we added the marks ('XD', 'DY', 'YX') of the driving routes in figure 11 and figure 12 (of the original manuscript) to indicate the driving direction. The telescope pointed backwards of the driving direction, which was illustrated in Section 2.1. We have amended the description of "vehicle flowrate was less" to "traffic flow was lower" in the revised manuscript.

Line 465: It is of note that there is a large positive offset on the TROPOMI HCHO. It appears that this offset is larger than the MAX-DOAS observed typical column. Discussion of the offset in addition to the correlation would be appropriate.

Reply: Many thanks for your comments. Previous studies found that the offsets of the TROPOMI HCHO were dependent on the HCHO concentration levels and presented to be positive at remote sites (Vigouroux et al., 2020). The larger positive offsets of the TROPOMI HCHO in this study were probably related to the HCHO horizontal inhomogeneity, caused by mountain terrains and varying local microclimates over the Tibetan Plateau. This discussion has been added in revised manuscript.

**Reference:**

Vigouroux, C., Langerock, B., Aquino, C. A. B., Blumenstock, T., Cheng, Z., Mazière, M. D., Smedt, I. D., Grutter, M., Hannigan, J. W., Jones, N., Kivi, R., Loyola, D., Lutsch, E., Mahieu, E., Makarova, M., Metzger, J.-M., Morino, I., Murata, I., Nagahama, T., Notholt, J., Ortega, I., Palm, M., Pinardi, G., Röhling, A., Smale, D., Stremme, W., Strong, K., Sussmann, R., Té, Y., Roozendael, M. v., Wang, P., and Winkler, H.: TROPOMI–Sentinel-5 Precursor formaldehyde validation using an extensive network of ground-based Fourier-transform infrared stations, Atmos. Meas. Tech., 13, 3751–3767, 10.5194/amt-13-3751-2020, 2020.

Figures 9 and 10: It would be useful for Figure 9 vertical axes to say that NO2 VCD is plotted, and for Figure 10 to say HCHO VCD on the axis.

Reply: Per your suggestion, the title of the vertical axes in Figure 9 and Figure 10 (of the original manuscript) has been amended as "$NO_2$ VCD" and "HCHO VCD", respectively.

---

## Author Comment (AC2)

**Referee #2**

Referee comments are in black. Author responses are in blue.

This paper presents mobile MAX-DOAS measurement of tropospheric nitrogen dioxide (NO2) and formaldehyde (HCHO) during summer months over the Tibetan plateau. Mobile MAX-DOAS made four closed loop journeys each spanning 3 days. Measurements of slant column densities (SCDs) at 15 degrees elevation angles (EA) are converted to vertical column densities (VCDs) using geometric approximation. The paper presents diurnal variation, and spatial variation of NO2 and HCHO VCDs in the Tibetan plateau. Using the terrain altitude of the drive track, it also presents the vertical profile of NO2 and HCHO VCDs over this remote background region. Finally, the measured NO2 and HCHO VCDs are used to validate TROPOMI measurements over the region. This paper provides a rare measurement over a data scarce region and hence is worthy of publication to ACP. However, major revision is needed, focused on characterizing the instrument detection limit and measurement uncertainty, and justifying some of the conclusions of the paper before it is accepted for publication.

Reply: First of all, we appreciate the reviewer's positive comments on our manuscript. In response to the reviewer's comments and suggestions, major revisions have been made in the revised manuscript. Listed below are our responses and the corresponding changes made to the manuscript according to the suggestions given by the reviewer.

**Major Comments:**

The main focus of the paper is providing measurements over a data scarce remote background region. However, the paper lacks discussion of the instrument detection limit and measurement uncertainty. Proper characterization of the detection limit and measurement uncertainty is very important so that the data presented in the paper are properly utilized in the future. Please include discussion of the instrument detection limit and measurement uncertainty. Based on the presented RMS values, most of the measurement appears to be close to or below the detection limits of the instrument. Please comment on the frequency of measurement at or below the detection limits, and how this impacts the reported background values for NO2 and HCHO of 4 x 1014 and 2.27 x 1015 molecule/cm2 respectively. Uncertainty due to geometric approximation also needs to be better characterized with some radiative transfer calculations and using measurements at different EA. Right now measurements at different EA are only being used to filter data. Absolute difference in VCDs between 15 and 20 EA of 1 x 1015 molecule/cm2 for NO2 and 2 x 1015 molecule/cm2 for HCHO is used as one of the filtering criterias. This is a factor of 1-2 higher than the mean background value so the measured VCDs could have error >100%.

Reply: Many thanks for your comments. Per your suggestions, we have added the discussion of instrument detection limit and measurement uncertainty in Section 3.1 of the revised manuscript. Based on the spectral fit errors corresponding to filtered NO2 and HCHO DSCDs, twice the medians of the spectral fit errors were estimated as the instrument detection limits for NO2 and HCHO, which are  $0.68 \times 10^{15}$  and  $2.11 \times 10^{15}$  molec cm-2, respectively. According to the DSCD detection limits divided by the DAMF for 15 ° elevation angle, the VCD detection limits were estimated to be about  $0.24 \times 10^{15}$  molec cm-2 for NO2 and  $0.74 \times 10^{15}$  molec cm-2 for HCHO, respectively. These values are very similar to the estimation of the background levels for NO2 and HCHO VCDs estimated by the maximum frequency method: the half widths at half maximum of the fitted curves were estimated to be their uncertainties ( $\pm 0.23 \times 10^{15}$  molec cm-2 for NO2 and  $\pm 0.96 \times 10^{15}$  molec cm-2 for HCHO), respectively (Section 4.1).

There are 17% and 15% of the retrieved  $NO_2$  and HCHO DSCDs below the detection limits, respectively. Based on the spectral fit errors, we can also calculate the relative errors for each  $NO_2$  and HCHO DSCD. Then the mean relative errors (uncertainties) of  $NO_2$  and HCHO DSCDs were about 21% and 12%, respectively.

It is a good suggestion to compare the geometric approximation and the atmospheric radiative transfer simulation. However, we lack necessary data during the field campaign to simulate the correct  $NO_2$  and HCHO AMFs. For example, the varying azimuth angle and shelter situation in these viewing direction are not known exactly along the driving routes. Thus we compare and filter  $NO_2$  and HCHO VCDs at different elevation angles, referring to previous study (Brinksma et al., 2008). In

this study, we used both absolute difference and relative difference as the filters, and data would be kept if at least one of both filters was fulfilled. The reason for choosing the absolute difference is to avoid to skip many measurements with low VCDs. With the condition of using two filters in this study, the means of the absolute differences and relative differences in the VCDs between  $15^{\circ}$  and  $20^{\circ}$  elevation angles are  $5.48 \times 10^{13}$  molec cm-2 and 11% for NO2 and  $3.02 \times 10^{14}$  molec cm-2 and 7% for HCHO respectively.

We also tested the applicability of the geometric approximation method by radiative transfer simulations using typical parameters. According to the AODs from the AERONET website (https://aeronet.gsfc.nasa.gov, last access: 2 December 2022) at three sites (Mt\_WLG, NAM\_CO, QOMS\_CAS) over the Tibetan Plateau, we estimate the AODs around 0.1 during our field campaign. But similar results are also found for AODs of 0.05 and 0.2. The simulation scheme is as the following.

| Parameters          | NO 2     | НСНО     |
|---------------------|---------------------|----------|
| Wavelength (nm)     | 440                 | 340      |
| Layer height (km)   | 0-1; 0-2            | 0-1; 0-2 |
| Aerosol height (km) | same as trace gases |          |
| AOD                 | 0; 0.05; 0.1; 0.2   |          |
| SZA()               | 20, 40, 60, 70, 80  |          |
| RAA()               | 10, 30, 60, 90, 180 |          |
| Elevation angle ( ) | 15                  |          |
| Terrain height (km) | 2, 3, 4, 5          |          |

The VCD ratios of the RTM simulations and the geometric approximation for 15 ° elevation angle under the condition of AOD=0.1 can be obtained for NO2 and HCHO, respectively. The DAMF ratios' means and standard deviations for all geometries (blue symbols) and RAA=10 °, SZA=60 ° and RAA=10 °, SZA=70 ° excluded (red symbols, for these rare measurement scenarios the strongest errors occur) are shown below. The main findings are that the typical errors of the geometric approximation are <20% for NO2 and HCHO.

---

## Author Response (AR1)

We thank the editor for handing this manuscript and the two anonymous referees for their insightful comments and constructive suggestions on our manuscript. Below are our point-to-point responses to the comments from each referee. Also enclosed is the revised version of the manuscript, marked up using revision track. Note that the Sections 4.2 (Temporal variation) and 4.3 (Spatial distribution) in the original manuscript were reorganized into Section 4.2 (Spatio-temporal variation) in the revised manuscript, suggested by Referee #2. This change is not marked up using revision track in order to keep the manuscript clear to read.

This manuscript describes mobile MAX-DOAS measurements recorded on drives around the Tibetan Plateau. The measurements are used to quantify the column of NO2 and HCHO along circular drive paths on the Plateau. The measurements are compared to TROPOMI satellite products. The manuscript represents a valuable contribution to the literature and reports ground truth around this relatively remote and high-altitude region. The manuscript uses the geometric method to retrieve tropospheric vertical column densities from differential slant column densities measured on the vehicle. This method, while lacking the refinement of one using radiative transfer calculations, is reasonable for this purpose. However there are some details on the method that should be discussed further and it would be valuable for other groups to understand how to optimize the measurements to get the most measurements from such studies. Concerns on the method are described below, followed by specific comments. If the concerns regarding the method can be addressed, this manuscript would be acceptable for publication in ACP.

Reply: First of all, we appreciate the reviewer's positive comments on our manuscript. In response to the reviewer's comments and suggestions, we have made relevant revisions to the manuscript. Listed below are our responses and the corresponding changes made to the manuscript according to the suggestions given by the reviewer. Note that the Sections 4.2 (Temporal variation) and 4.3 (Spatial distribution) in the original manuscript were reorganized into Section 4.2 (Spatio-temporal variation) in the revised manuscript. This change is not marked up using revision track in order to keep the manuscript clear to read.

Were the elevation angles recorded relative to the mobile vehicle or to gravity? If they were gravitationally referenced, was a gyroscope used? If they are relative to the road, how is the local horizon taken into consideration? I think lines 260-265 indicate that the angles are with respect to the vehicle, and that is a reason to use

higher elevation angles, but the text is not very clear to this regard. Can the authors be more clear about how the view geometry is defined?

Reply: The elevation angles were recorded relative to the mobile vehicle. We didn't use a gyroscope. During our deployment strategies, we designed a partial system recording the attitude angles of the mobile vehicle to correct the elevation angle of MAX-DOAS measurements. However, it did not work well and couldn't be used for this study. To reduce the influences of local non-horizontal road on the un-corrected elevation angle, we used the DSCDs at larger elevation angles. Also, the VCDs were further filtered based on the absolute difference and the relative difference of VCDs between 15 ° and 20 °. The definition of elevation angle has been added to the text in Section 2.1 of the revised manuscript.

As is pointed out by the authors, the higher altitude and lower aerosol extinction conditions of a sparsely populated plateau make the geometric approach more tenable. However, clouds or aerosol-particle-rich pollution plumes (which may accompany NO2) may affect this assumption. Therefore, it seems reasonable to check this assumption by using O4 observations. The authors could calculate the O4 VCD above the vehicle from the altitude (which they know) and pressures at meteorological stations (or better soundings if available). They can then use the mobile-measured geometrically calculated O4 VCD\_15 to compare to the meteorologically calculated one. Due to radiative transfer effects, particularly the relative azimuth angle to the sun, these quantities won't be perfect, but times when there is a large amount of aerosol or clouds or the view azimuth happened to be close to the sun, one could tell that the NO2 and HCHO data are being affected by these confounding effects. This check should help to assure that the geometric method is working for NO2 and HCHO.

Reply: Many thanks for your suggestions. The procedures of O4 VCDs derived from MAX-DOAS measurements and calculations are as the following.

(1) The O4 DSCDs were retrieved from MAX-DOAS spectra in the wavelength interval of 351-390 nm using a sequential FRS and are then filtered by the conditions of SZA <80 °, RMS < 0.005, offset (constant) between  $\pm 0.03$ . The O4 VCDs obtained by the geometric approximation method at 15 ° elevation angle were further filtered by the differences of the O4 VCDs between 15 ° and 20 °. The O4 VCDs were kept if the

absolute difference of VCDs between  $15^{\circ}$  and  $20^{\circ}$  is  $< 1 \times 10^{42}$  molec2 cm-5 or the relative difference is <10%.

(2) During the procedure of the O4 VCD calculation, we used the air temperature and pressure profiles of hourly ERA5 with 0.25 °×0.25 ° grid above the altitude of the driving route. Firstly, we extracted the profiles of temperature and pressure matched with each measurement at the same grid cell and the same hour. Then we calculated the O2 concentrations from the surface to 30 km at each altitude with a vertical interval of 50m. The O4 concentrations were assumed as the square of the O2 concentrations at each vertical grid cell and integrated as O4 VCDs from the surface to 30 km (Wagner et al., 2019).

The results of  $O_4$  VCDs derived from the MAX-DOAS measurements and calculated from ERA5 data are shown in the figure below. The main finding is that the measured  $O_4$  VCDs are systematically lower than the calculated ones. Part of the underestimation is probably related to clouds, but a strong underestimation is also found for measurements for clear skies.

Hence we further explored the applicability of the geometric approximation method by radiative transfer simulations. According to the AODs from the AERONET website (https://aeronet.gsfc.nasa.gov, last access: 2 December 2022) at three sites (Mt\_WLG, NAM\_CO, QOMS\_CAS) over the Tibetan Plateau, we estimate the AODs

| Parameters          | O 4         | NO 2     | НСНО              |
|---------------------|------------------------|---------------------|-------------------|
| Wavelength (nm)     | 340                    | 440                 | 340               |
| Layer height (km)   | US standard atmosphere | 0-1; 0-2            | 0-1; 0-2          |
| Aerosol height (km) | 0-1                    | same as trace gases |                   |
| AOD                 | 0; 0.05; 0.1; 0.2      | 0; 0.05; 0.1; 0.2   | 0; 0.05; 0.1; 0.2 |
| SZA()               | 20, 40, 60, 70, 80     |                     |                   |
| RAA()               | 10, 30, 60, 90, 180    |                     |                   |
| Elevation angle ( ) | 15                     |                     |                   |
| Terrain height (km) | 2, 3, 4, 5             |                     |                   |

around 0.1 during our field campaign. But similar results are also found for AODs of 0.05 and 0.2. The simulation scheme is as the following.

The VCD ratios of the RTM simulations and the geometric approximation for 15 ° elevation angle under the condition of AOD=0.1 can be obtained for O4, NO2, and HCHO, respectively. The DAMF ratios' means and standard deviations for all geometries (blue symbols) and RAA=10 °, SZA=60 ° and RAA=10 °, SZA=70 ° excluded (red symbols, for these rare measurement scenarios the strongest errors occur) are shown below. The main findings are: (1) The typical errors of the geometric approximation are <20% for NO2 and HCHO; (2) The errors of the geometric approximation are much larger for O4 with a systematic underestimation between about 40% and 60%, which are in overall agreement with the comparison of the measured and calculated O4 VCDs above; (3) The large underestimation of the O4 VCDs indicates that O4 can not be used for the test if the geometric approximation is justified or not for an individual measurement of NO2 and HCHO.

---

## Author Response (AR2)

We thank the editor for handing this manuscript and the two anonymous referees for their insightful comments and constructive suggestions on our manuscript. Below are our point-to-point responses to the comments from each referee. Also enclosed is the revised version of the manuscript, marked up using revision track. Note that the color schemes in Figures 8, 11, S7 have been adjusted in the revised manuscript which allow readers with color deficiencies to correctly interpret our findings, as suggested by the editorial support team.
Thanks to the authors for responding to the prior reviews. These were useful checks, but I think that these have led to some aspects that deserve further exploration. Specifically, the prior reviews requested that the authors check the geometric approximation using O$_4$ data, consider ground slope, further address the signal to noise of the observations, and consider comparisons to satellite observations. Below are some aspects of these issues that should be explored by the authors such that a revised manuscript can be made and re-considered for publication.

Reply: First of all, we appreciate the reviewer's comments on our manuscript. In response to the reviewer's comments and suggestions, we have made relevant revisions to the manuscript. Listed below are our responses and the corresponding changes made to the manuscript according to the suggestions given by the reviewer.

- Failure of geometric model to retrieve O$_4$ VCD:

The response to reviewers indicates that O$_4$ is not well retrieved by the geometric approximation using 15 ° elevation angle viewing geometry. The O$_4$ VCD retrieved using the geometric approximation is often half (~40 to 60%) of the column that is calculated from meteorological data. This is worrisome with respect to the quantification of VCDs by the geometric method. The authors have made a useful set of calculations using a radiative transfer model (RTM), which appears to show that this underrepresentation of O$_4$ is also in the RTM. The authors do not discuss why O$_4$ should fail but the geometric approximation should succeed for NO$_2$ and HCHO, despite their desire to keep using the geometric method for NO$_2$ and HCHO. The authors should explore for reasons why O$_4$ fails by the geometric method yet apparently succeeds for NO$_2$ and HCHO in order to keep using the geometric method for these gases.

One possible speculation could be that O$_4$ at larger heights above the surface is not contributing as much to the SCD as does O$_4$ nearer the surface. The scale height of O$_4$

is about 3.5km (half the scale height of pressure due to $O_4$ amount being proportional to the square of $O_2$). The table in the reply to reviewers indicates that $NO_2$ and HCHO are assumed to have a layer height of 0-1km or 0-2km (I presume above the terrain), so even with a 2km thick layer of these gases, most of the $O_4$ column is above the 2km top of these layers. If the geometric approximation is less sensitive to this higher-altitude part of the $O_4$ column, it might explain why the geometric approximation is failing for $O_4$. This hypothesis could be tested by splitting the $O_4$ distribution in the RTM into a below 2km (AGL) and an above 2km part and running the model on these two parts. Whether this idea proves out or not, the authors should explore and discuss reasons why the geometric approximation for $O_4$ failed yet they want to keep using it for $NO_2$ and HCHO.

Reply: As suggested by the reviewer, we performed additional RTM simulations. We considered the following scenarios:

a) $O_4$ profile with the concentration in the lowest 2 km set to zero

b) $O_4$ profile with the concentration above 2 km set to zero

c) HCHO background profile (Fig. S2) with the concentration in the lowest 2 km set to zero

d) $O_4$ profile with effects of clouds with an optical depth of 10 at different altitudes (2-3km, 4-5km, 8-9km)

e) Trace gas box profile from the surface to 2km with effects of clouds with an optical depth of 10 at different altitudes (2-3km, 4-5km, 8-9km)

The results are shown in the new figures S3 and S4, see below.

The main conclusions are:

1) the sensitivity of the geometric approximation for $O_4$ is high (~90%) for the part below 2km, but is low (~40%) for the part above 2km. This explains why the $O_4$ VCD retrieved using the geometric approximation is generally underestimating the true $O_4$ VCD.

2) the sensitivity of the geometric approximation for the part of the background HCHO above 2km is also low (~40%). Thus the use of the geometric approximation systematically underestimates the background HCHO above 2km. However, from model simulations over the Tibetan Plateau (Fig. S2), we find that the corresponding vertical HCHO column density is rather small: about $1.3 \times 10^{15}$ molec/cm² Thus the retrieved HCHO VCDs underestimate the true total VCD by about $0.6 \times 1.3 \times 10^{15}$ molec/cm² $\approx 8 \times 10^{14}$ molec/cm²

3) in the presence of clouds, the sensitivity for trace gases below the clouds is slightly enhanced compared to the geometric approximation, while it is strongly reduced for trace gases above the clouds. This finding confirms the assumption that clouds lead to a further underestimation of $O_4$ (and background HCHO) while the sensitivity for the trace gases close to the surface is almost unchanged.

An important reason to explore the failing of the geometric method for $O_4$ is that for smaller columns of HCHO, there is a larger contribution of "background" HCHO arising from oxidation of methane to the HCHO VCD. This "background" HCHO is not just in the boundary layer, but extends further aloft because methane is fairly well mixed in the troposphere. Therefore, the actual profile of HCHO might not be the assumed 0-1km or 0-2km layer, which succeeded in the geometric approximation retrieval, but might look more like that of $O_4$, and might therefore be underestimated by the geometric approximation (as O4 is). Figure S1 in the revised supplement may start to hint at increased underestimation of the true column for HCHO as the layer thickness increases from 0-1km to 0-2km. I believe that some satellite retrievals measure the differential VCD compared to a reference sector and then add back this "background" column of HCHO to get a total column. The "background" column typically comes from a global chemical transport model and examination of the column over this high-altitude region could give an estimate of the fraction of the HCHO column that is in the background (and thus potentially under-represented in the geometric retrieval). Figure S2 in the revised supplement indicates that HCHO's mixing ratio profile is much more constant with altitude than $NO_2$, which may indicate that the HCHO concentration profile extends to higher altitudes AGL than does the $NO_2$ profile, potentially indicating that the RTM calculations that assume HCHO is in the 0-1km (AGL) or 0-2km layer are not appropriate. The authors should explore how their geometric method would work for "background HCHO" and use satellite / GCM estimates of the background to determine how much of the columns they are observing may be not in the boundary layer.

Reply: Many thanks for this good suggestion! We made the corresponding RTM simulations (see also above) using a typical HCHO background profile (Fig. S2) from a GCM (Ma et al., 2019). We found that the sensitivity of the geometric approximation for the background HCHO above 2 km is only about 40%, leading to a systematic underestimation of the total HCHO VCD by about $8 \times 10^{14}$ molec/cm².

**Reference:**

Ma, J., Brühl, C., He, Q., Steil, B., Karydis, V. A., Klingmüller, K., Tost, H., Chen, B., Jin, Y., Liu, N., Xu, X., Yan, P., Zhou, X., Abdelrahman, K., Pozzer, A., and Lelieveld, J.: Modeling the aerosol chemical composition of the tropopause over the Tibetan

Plateau during the Asian summer monsoon, Atmospheric Chemistry and Physics, 19, 11587-11612, 10.5194/acp-19-11587-2019, 2019.

In the reply to reviewers, the authors say: "Part of the underestimation is probably related to clouds, but a strong underestimation is also found for measurements for clear skies." It would be useful to give more information on this statement. Specifically, there are times when the $O_4$ VCD is very small (e.g. on July 25, 2021) during a cloudy period? Can the authors indicate when there were clouds on their timeseries so that we can understand the effect of those clouds? Although the author's radiative transfer simulations can help to address questions of largely clear-sky behavior (they have AOD up to 0.2), the simulations do not help address understanding of cloudy behavior. It may be the case that if there is a cloud that is above the boundary layer $NO_2$ that the presence of the cloud might not affect the retrieval much, but the authors have not shown that. Can the authors expand the radiative transfer simulations to include a layer cloud aloft? That seems like a situation that should be addressable with their model.

Reply: Many thanks for your comments.

We performed additional RTM simulations for cloudy situations (clouds with an OD of 10 at different altitudes: 2-3km, 4-5km, 8-9km).

As expected, the sensitivity for the trace gases below the clouds is hardly affected (even slightly enhanced), but for trace gases above the cloud it is strongly reduced. These findings can explain the observed strong underestimation of the $O_4$ VCDs for some days.

It is a pity that we have no detailed information on the time series of clouds along the driving routes. We only qualitatively estimate the cloud conditions based on the observer manual recordings. As a whole, the weather conditions are dominated by the sunny sky and light rain or cloudy sky for the second and third circling journeys, respectively. And yes, it is cloudy on July 25, 2021.

Although the radiative transfer model calculations are useful, details on these calculations are lacking. For example, which radiative transfer model is used? Presumably some aerosol properties (e.g. asymmetry factor, single scattering albedo) are used, but are not stated. The $O_4$ simulations say 0-1000m in their caption box on the figure, which I guess is the aerosol layer thickness because $O_4$ goes a lot higher than that. Please clarify what this height range refers to.

Reply: We added the following information to the revised version of the paper:

-the Monte Carlo RTM MCARTIM was used for the simulations (see Deutschmann et al., 2011).

-the aerosol optical properties are: asymmetry parameter: 0.68, single scattering albedo: 0.95

-the cloud optical properties are: asymmetry parameter: 0.85, single scattering albedo: 1.0

-the altitude information in the figures describes the layer height of the aerosols, and for $NO_2$ and HCHO also the trace gas layer heights.

**Reference:**

Deutschmann, T., S. Beirle, U. Frieß, M. Grzegorski, C. Kern, L. Kritten, U. Platt, C. Prados-Román, J. Puķı̄te, T. Wagner, B. Werner, K. Pfeilsticker, The Monte Carlo atmospheric radiative transfer model McArtim: Introduction and validation of Jacobians and 3D features, Journal of Quantitative Spectroscopy and Radiative Transfer, Volume 112, Issue 6, 1119-1137, https://doi.org/10.1016/j.jqsrt.2010.12.009, 2011.

● Effect of ground slope:

The authors calculate that errors up to 21% can arise from slope, which is a good number to keep in mind. The authors then go on to average positive and negative slope errors to get a near zero error (1%). However, that calculation assumes equal mix of up and downhill driving, while in fact there might not be an equal mixture. Later they indicate that only about 1 minute in 8 minutes is observing at 15°, so the slope during that period is what matters, and driving up or down a slope for a minute seems very reasonable in an area that covers ~3km vertical range. I think it would be safer to say that the ground slope may lead to an error of +/-21%, but that over the full loop these errors should at least partially cancel. It is possible that these errors might contribute to the low correlation between the mobile measurements and satellite observations.

Reply: Many thanks for your comments. The descriptions about the effect of the ground slope in the revised manuscript have been amended as: "The corresponding error of an individual measurement will be up to about 21%, but over the full loop

these errors will at least partially cancel.".

In addition, although on the duration of the measurement at 15° elevation angle is only 1 min, we compared VCDs derived for different elevation angles (at different times of the same measurement sequence) and then the $VCD_{15°}$ were filtered based on the differences of VCDs between 15° and 20°, which can partially eliminate the VCDs with large errors when there are significant differences in ground slope between different elevation angles.

Yes, the errors will affect the correlation between the mobile measurements and satellite observations. However, for the comparison with TROPOMI observations, we use the means of typically 2-3 data points at a specific grid (0.25°×0.25°) in order to reduce the uncertainties of the VCDs from both mobile MAX-DOAS and TROPOMI.

- Error estimates:

Text was added to the end of section 3.1 describing error analysis. The authors appear to use two times the median spectral fit error. I presume the fit error is like a standard deviation (sigma), so this is 2*sigma, a reasonable definition of detection limit, but the text should be more clear. These (2-sigma) DL are $0.24×10^{15}$ molecule cm^-2 for NO2 and $0.74×10^{15}$ molecule cm^-2 for HCHO. These detection limit estimates use the airmass factor at 15 ° (e.g. the geometric approximation), but no error is added for uncertainties in the geometric approximation. I think that at least 21% error for road tilt and ~20% error from the radiative transfer calculations should be added to this spectroscopic-only error estimate.

Reply: Many thanks for your comments. During DOAS measurements, the detection limit can be conveniently estimated by the spectral fit errors (Cheng et al., 2021; Coburn et al., 2011; Stutz and Platt, 1996). We clearly stated the definition of the instrument detection limit in the revised manuscript, which is traceable for future references.

Due to the detection limit estimated from the medians of the DOAS fit errors over the full loop (including the conditions of driving up and down a slope), the error for the road tilt is probably much smaller than 21% and not the main error among the error sources of detection limit. Nevertheless, in the revised version of the manuscript,

we added the following sentence to mention the effects of the geometric approximation and the ground slope:

'Note that for individual measurements, the VCD detection limits might be lower or higher by about ±30% because of the uncertainties of the geometric approximation (up to about 20%) and the effect of varying ground slope (also up to about 20%).'

In section 4.1 (and the abstract), "background" levels of these gases are described, with an +/- listed (I would have assumed to be an error estimate), yet it is differently defined than the section 3.1 error analysis. In section 4.1, the text says "The uncertainties of the background levels were estimated by the half width at half maximum of Lorentz fitted curves (Fig. 6a)." I think that these are not "uncertainties", but rather the combination of variability in the species combined with analytical uncertainties in the measurements. The Lorentzian half width is used, which I think would be narrower than 1-sigma of a Gaussian fit. Can the authors justify why a Lorentzian is used here rather than a Gaussian? The Gaussian is connected to normal statistical error analysis and seems preferable, although the distributions do look longer tailed than a Gaussian. Overall, the similarity of section 3.1 detection limits and the width of the distributions in Figure 6 would lead me to believe that a significant part of the width of these distributions is instrumental noise. The quoting of the Lorenzian half width in the abstract seems misleading to me as I would have expected that the +/- number listed would be a Gaussian error estimate, possibly even

2-sigma. Please clarify this error discussion and make sure that the definition of the error estimate is included in the abstract.

Reply: Many thanks for your comments. The description about the background level and its uncertainties has been amended in the revised manuscript: "According to the Lorentz fitted curves of the relative frequency distribution of the $NO_2$ and HCHO VCDs during the field campaign (Fig. 6a), the background levels were $0.40 \pm 1.13$ $\times 10^{15}$ molec·cm$^{-2}$ for $NO_2$ and $2.27 \pm 1.66 \times 10^{15}$ molec·cm$^{-2}$ for HCHO in summer on the northeast of the Tibetan Plateau. Wherein the uncertainties of the background levels were estimated by the standard deviations of the $NO_2$ and HCHO VCDs". The corresponding revisions have been made in the sections of abstract and conclusions.

The purpose of curve fitting is to find the peaks of relative frequency distribution of the $NO_2$ and HCHO VCDs during the field campaign. The bigger relative frequencies are concentrated near the peak, meaning that the curve peak fitting should use the function with relatively narrow line width. Therefore, we prefer to use the Lorentz function, which fits the relative frequency distribution of the $NO_2$ and HCHO VCDs well in fact (Fig. 6a).

● Comparison to satellite:

The text says "Interestingly, there is almost no correlation of the two data sets, if we only use the tropospheric $NO_2$ VCDs within the 1.5 h time difference between mobile MAX-DOAS and TROPOMI at the same grid (referred to 'ΔT1.5' in Fig. 15a, corresponding to the red pluses in Fig. 13)." It seems to me that the noise on the measurements, both satellite and ground based, coupled with effects like slope, variable solar geometry, etc. are all going to reduce the correlation between the two data sets, particularly due to the low levels of these pollutants at these high-altitude remote sites. Therefore, I think the weak correlation is to be expected and is a product of the low level of pollution. Although the correlation is poor, the difference on average of the data by both methods (the bias) is a useful result of the study. I believe that the other reviewer also seeks to have greater focus on the bias in measurements than correlation (given noise on both measurements). The discussion of this correlation plot should include reference to the error estimates discussed above and also should discuss errors on TROPOMI measurements.

Reply: Many thanks for your comments. Yes, the weak correlation is to be

understandable: (1) The level and the range of variation of the $NO_2$ VCDs are very small in the background atmosphere over the Tibetan Plateau; (2) The signal-to-noise ratio is reduced due to the measurement errors for both MAX-DOAS (see section 3) and TROPOMI, introduced by the spectral analysis, ground slope, and the applied tropospheric AMF. The TROPOMI relative precisions in the '$\Delta T_{1.5}$' situation are estimated to be 72% and 113% for tropospheric $NO_2$ and HCHO VCDs, derived from the products of S5P_L2__NO2____HiR and S5P_L2__HCHO___HiR, respectively. These information has been added in the revised manuscript.

We have added the absolute differences (i.e. bias VCD values) in the revised manuscript. The varying mountainous terrain could lead to the horizontal inhomogeneity of the $NO_2$ and HCHO VCDs. The satellite measurements represent the averaged $NO_2$ and HCHO VCDs at a specific grid cell, while MAX-DOAS observations reflect the $NO_2$ and HCHO VCDs in a specific viewing direction. Therefore, one possible reason for the VCD bias between the two methods is the effect of mountainous terrain. However, without detailed knowledge about the true 3D $NO_2$ and HCHO distribution, this bias can not be fully understood in direction and magnitude. A corresponding description has been amended in the revised manuscript.
Reply: First of all, we appreciate the reviewer's positive comments on our manuscript. Listed below are our responses and the corresponding changes made to the manuscript according to the suggestions given by the reviewer.

DOAS detection limit is often defined based on the RMS noise than DOAS fit error (e.g. Stutz and Platt, 1996, Coburn et al., 2011). 1 σ RMS roughly corresponds to 3 σ DOAS fit error. The authors have conveniently decided to use 2 σ DOAS fit error as the detection limit which results in most the measurements being above the detection limit. The authors have clearly stated their definition of the detection limit so I think it is fine and traceable for future references.

Stutz, J. and Platt, U.: Numerical analysis and estimation of the statistical error of differential optical absorption spectroscopy measurements with least-squares methods, Appl. Opt., 35, 6041– 6053, doi:10.1364/AO.35.006041, 1996.

Coburn et al.: The CU ground MAX-DOAS instrument: characterization of RMS noise limitation and first measurements near Pensacola, FL of BrO, IO and CHOCHO.

Reply: Many thanks for your comments. With respect to the methods of obtaining the DOAS detection limit, the information above is important. We have added the two references (i.e. Stutz and Platt, 1996, Coburn et al., 2011) to the revised manuscript, so that the paper will be more readable and referable.

It is not clear how mountainous terrain would result in positive NO₂ and HCHO bias for TROPOMI. I suggest the authors also add bias VCD values in the paper along with the % bias.

Reply: Many thanks for your comments and suggestions.

[revised manuscript text omitted]

840 **Figure 11.** Same as figure 8, but for HCHO.

[Figure]

**Figure 12.** Same as figure 9, but for HCHO.

[Figure]

**Figure 13.** Spatial distributions of the tropospheric NO₂ VCDs observed by TROPOMI on each day of the field campaign. The TROPOMI S5P_L2__NO2____HiR product has been gridded to 0.25 °×0.25 ° cells. The main cities and counties on the driving routes of the field campaign are marked by the black stars. The black curves indicate the administrative boundaries. The white circles and red plus symbols show the grid cell where the data of both TROPOMI and MAX-DOAS are available on the same day or within a 1.5 h time difference, respectively.

[Figure]

850

**Figure 14.** Same as figure 13, but for HCHO.

[Figure]

[Figure]

**Figure 15.** Linear fit between the tropospheric **(a)** $NO_2$ and **(b)** HCHO VCDs measured by the mobile MAX-DOAS and TROPOMI. The black squares and red dots represent the available VCDs of both data sets at the same grid cell on the same day or within a 1.5 h time difference, respectively. The black (red) lines denote the results of the regression analyses and the corresponding equations and correlation coefficients are displayed in the figures.

855